# Chemoorganoautotrophic lifestyle of the anaerobic enrichment culture N47 growing on naphthalene
Isabelle Heker [1], Christian Seitz[2], Lisa Voskuhl [1], Yachao Kong[1], Isabell Erdmann[1], Frederik Götz[1], Mohamed Hassoun[1], Claudia Huber [2], Wolfgang Eisenreich [2] & Rainer U. Meckenstock [1] ✉

In almost all respiratory organisms, organic substrates are degraded via catabolic processes to the central metabolite acetyl-CoA which is then oxidized to $CO_2$ for energy metabolism or used as a building block for anabolism. Most microorganisms have either the closed tricarboxylic acid cycle or the complete Wood-Ljungdahl-pathway for acetyl-CoA oxidation, but the sulfate-reducing, naphthalene-degrading culture N47 possesses both completely. Combining $^{13}C$- labeled substrates and mass-specific GC-MS analysis of amino acids and fatty acids with enzyme activity assays suggests that N47 has a chemoorganoautotrophic metabolism degrading complex organic substrates such as naphthalene. Surprisingly, however, the biomass is mainly produced from acetyl-CoA generated de novo via $CO_2$-fixation. This metabolism probably requires both a complete Wood-Ljungdahl pathway for acetyl-CoA oxidation and a reverse tricarboxylic acid cycle for $CO_2$ fixation. Based on genome analysis, this chemoorganoautotrophic metabolism seems to also occur in other sulfate-reducers and anaerobic ammonium-oxidizers.

Most respiratory organisms feeding on organic compounds are chemoorganoheterotrophic, meaning that they use chemical reactions for energy conservation (chemotrophy) as opposed to using light energy (phototrophy); they use organic substrates as electron source rather than inorganic compounds (organotrophy vs. lithotrophy), and organic carbon is used for biomass formation rather than $CO_2$-fixation (heterotrophy vs. autotrophy). A central metabolite in the oxidation of organic compounds is acetyl-CoA, which can be further oxidized to $CO_2$ for energy conservation by for example the tricarboxylic acid (TCA) cycle, which is commonly used in various different organisms, or the Wood–Ljungdahl pathway (WLP), which occurs only in anaerobic microorganisms. In anabolism of heterotrophic organisms, that is organisms using organic carbon for biomass formation, acetyl-CoA is used to synthesize building blocks such as fatty acids, amino acids, sugars, and others. This is favored over autotrophic growth that relies on fixation of inorganic $CO_2$ to produce biomass, because it requires fewer reducing equivalents and energy than $CO_2$-fixation. The only known exception to chemoorganoheterotrophic organisms feeding on organic compounds refers to methylotrophic microorganisms. Among these, ANME archaea (anaerobic methanotrophic archaea) oxidize their C1-substrate methane to $CO_2$ and produce their lipids by assimilation of inorganic carbon[1]. However, these organisms do not produce acetyl-CoA in

the methane oxidation pathway, which excludes using the energy-rich compound to be directly used for anabolism. The fact that these organisms conserve energy from the degradation of simple organic substrates like methane to $CO_2$, making them chemoorganotrophs, but at the same time they fix $CO_2$ for biomass formation (autotrophy), leads to the term "chemoorganoautotrophy" for this type of metabolism[1]. For reviews on fundamentals of anaerobic metabolism and energy conservation refer to Thauer et al., 1978, 1988, and 1989[2–4], and about anaerobic degradation of (aromatic) hydrocarbons read Meckenstock et al.[5] and Rabus et al.[6].

Here, we demonstrate that chemoorganoautotrophic metabolism (chemical reactions for energy conservation, organic electron source, and $CO_2$ fixation as carbon source) is not restricted to methylotrophs only, but can also occur in microorganisms that are degrading more complex organic substrates such as polycyclic aromatic hydrocarbons (PAHs). We provide evidence for this metabolism in anaerobic, sulfate-reducing microorganisms that degrade polycyclic aromatic hydrocarbons which are hazardous and widespread pollutants in the environment[7–9]. Anaerobic degradation of polycyclic aromatic hydrocarbons is well known for the highly enriched, sulfate-reducing culture N47 which degrades naphthalene as carbon and electron source with sulfate as electron acceptor[10–14] and mainly consists of *candidate* Desulfobacterium strain N47 (99.6% relative abundance). In this

[1]Faculty of Chemistry, Environmental Microbiology and Biotechnology, University of Duisburg-Essen, Essen, Germany. [2]Bavarian NMR Center - Structural Membrane Biochemistry, Department of Bioscience, School of Natural Sciences, Technical University of Munich, Garching, Germany. ✉e-mail: rainer.meckenstock@uni-due.de

 1

culture, naphthalene is activated by carboxylation[11] and, after ligation with coenzyme A, naphthoyl-CoA[15] is reduced by two novel type III aryl-CoA reductases[16–18] and a type I aryl-CoA reductase[13,17]. Then, beta-oxidation like enzymes open the aliphatic ring system finally leading to the central metabolite acetyl-CoA.

Anaerobic microorganisms can oxidize acetyl-CoA via different pathways[19–21], but usually have either the TCA cycle or the WLP. Most organisms that use the WLP additionally use parts of the TCA for providing building blocks for e.g., amino acid synthesis but they do not have a closed TCA cycle. The missing key enzyme is here 2-oxoglutarate dehydrogenase which connects the two branches of a complete TCA. When the proteome and metagenome of *candidatus* Desulfobacterium strain N47 were analyzed, it surprisingly reflected that most genes of a complete TCA cycle and a complete WLP were both abundant and also expressed in the N47 culture grown with naphthalene[22]. The aconitate hydratase, also known as aconitase, was identified only in the N47 metagenome but not in the proteome and some subunit genes (2-oxoglutarate:ferredoxin oxidoreductase subunit delta, succinate dehydrogenase flavoprotein subunit, succinate dehydrogenase iron-sulfur subunit, succinate dehydrogenase cytochrome *b* subunit) were not found in the metagenomic bin but in the unbinned metagenome and in the proteome. Additionally, all other subunits of these enzymes could be identified in the bin and their absence is likely due to limitations in the bioinformatics pipeline rather than their absence in the organism's genome. The bin's completeness of only 72.90 and 0% contamination, also suggests that genes were missed during assembly or binning (Table S1). Hence, both WLP and TCA are complete.

Most of the enzymes involved in the regular (oxidative) TCA cycle are reversible. Only the reactions catalyzed by citrate synthase and 2-oxoglutarate dehydrogenase were considered for a long time to function only in the forward (oxidative) direction. However, using the same set of enzymes, a few organisms can use this cycle also in the reductive direction under very special conditions, such as high $CO_2$ partial pressure and elevated levels of citrate synthase in the cell, though the enzyme activities in this direction are lower than in the oxidative direction. This pathway is referred to as the "reversed oxidative TCA cycle" (roTCA)[21,23]. Specifically, it was discovered that the thermophilic sulfur-reducing deltaproteobacterium *Desulfurella acetivorans* and *Hippea maritima* can drive this roTCA cycle when high levels of $CO_2$ are present. Experiments demonstrated faster growth for *H. maritima* with generation times around 17 h at 20% and 40% $CO_2$ in the gas phase, compared to slower growth (29 h generation time) at 5% $CO_2$ and no growth at 1 and 2% $CO_2$ in the gas phase[21,23]. However, this route also required high levels of citrate synthase and a ferredoxin-dependent 2-oxoglutarate synthase instead of the NAD-dependent 2-oxoglutarate dehydrogenase[23]. Several other microorganisms use the TCA cycle in the reductive direction (rTCA cycle) for $CO_2$ fixation by replacing the few enzymes that catalyze virtually irreversible steps in the oxidative TCA cycle. For example, in the rTCA cycle, 2-oxoglutarate:ferredoxin oxidoreductase and ATP citrate lyase catalyze fully reversible reactions instead of the virtually irreversible 2-oxoglutarate dehydrogenase and citrate synthase of the oxidative TCA cycle[2,24–26]. Notably, both the WLP and the TCA cycle can theoretically run in both directions, i.e., for acetyl-CoA oxidation to $CO_2$ or $CO_2$ fixation to acetyl-CoA, respectively.

In this study, we analyzed the carbon flow in culture N47 and determined which pathways are involved in its metabolism. We found indications for both an active TCA cycle as well as for a WLP. $^{13}C$-Labeling experiments with N47 and enzyme activities in cell-free extracts of N47 suggested that N47 employs a chemoorganoautotrophic metabolism of (i) degrading naphthalene to $CO_2$ for energy conservation and (ii) assimilating $CO_2$ for biomass formation at the same time. This would make N47 the first known organism to use this kind of metabolism while growing on a complex organic substrate like naphthalene. We hypothesize that this metabolism can be performed by a combination of the WLP for acetyl-CoA oxidation and a reverse TCA for fixation of $CO_2$.

## Results and discussion
### Enzyme activities of TCA and WLP
To analyze if the enzymes of both the TCA cycle and the WLP were fully active in cell-free extracts of culture N47, cells were grown with naphthalene

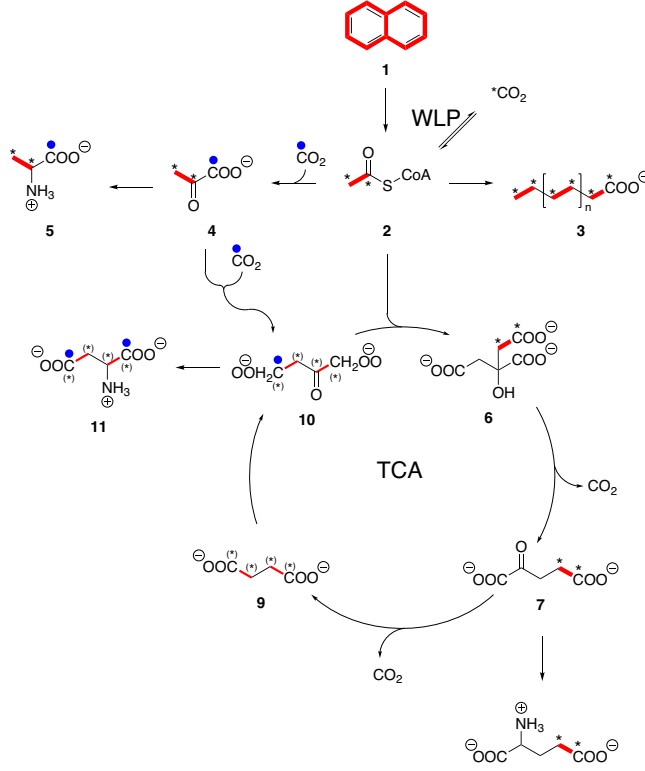

**Fig. 1 | Overview of the metabolic pathways of fatty acids and some amino acids.** Thick red lines indicate adjacent $^{13}C$-atoms (e.g., $^{13}C_2$ bodies, two labeled carbon atoms in the same molecule). Blue dots indicate $^{13}C$-atoms derived from $^{13}CO_2$ (one labeled carbon atom). $^{13}CO_2$ can react with $^{12}CO_2$ in the WLP to form acetyl-CoA with a mass increase of one, marked with an asterisk *($C_2$ body, one labeled carbon atom). Due to the symmetry of succinate, it is not possible to observe if the label is on C1 and C2 or C3 and C4, respectively, which is indicated with a thinner red line and (*). **1**: naphthalene, **2**: acetyl-CoA, **3**: exemplary fatty acid, **4**: pyruvate, **5**: alanine, **6**: citrate, **7**: 2-oxoglutarate, **8**: glutamate, **9**: succinate, **10**: oxaloacetate, **11**: aspartate.

**Table 1 | Enzyme activities of the TCA cycle and WLP measured in cell-free extracts in oxidative and reductive direction by spectrophotometric assays and liquid chromatography-MS (LC-MS)**

| Enzyme | TCA cycle | | WLP |
|---|---|---|---|
| | forward | reverse | |
| Citrate synthase | 2.0 ± 0.7 | 0.05 ± 0.008 | |
| Aconitase | 3.6 ± 0.42* | | |
| Isocitrate dehydrogenase | | 65 ± 9.5 | |
| 2-oxoglutarate:ferredoxin oxidoreductase | 1.1 ± 0.20 | | |
| Succinate dehydrogenase | 70 ± 15 | 9.4 ± 0.58 | |
| Fumarase | 0.15 ± 0.01 | 1.1 ± 0.46 | |
| Malate dehydrogenase | 0.88 ± 0.24 | 2.9 ± 0.51 | |
| Carbon monoxide dehydrogenase | | | 94 ± 14 |
| Formate dehydrogenase | | | 440 ± 83 |

All activities are given in nmol min$^{-1}$ mg$^{-1}$ protein, errors depict measurement uncertainties of three replicate enzyme assays performed from one batch of cell free extract, taking into account the error of the protein concentration measurements that were also carried out in triplicates. Values marked with an asterisk (*) represent activities that could only be measured in combined enzyme assay reactions.

**Table 2 | Excess of M+1 and M+2 isotopologues in amino acids and fatty acids obtained from culture N47 grown with either H$^{13}$CO$_3^-$/$^{13}$CO$_2$ (10 atom% $^{13}$C) or [U-$^{13}$C$_{10}$]naphthalene (10 atom% $^{13}$C)**

| | Mass of fragment (M) | [U-$^{13}$C]naphthalene | | $^{13}$CO$_2$ | |
|---|---|---|---|---|---|
| | | M+1 (%) | M+2 (%) | M+1 (%) | M+2 (%) |
| Alanine | 260 | 2.4 (±0.4) | 0.4 (±0.2) | 7.8 (±0.3) | <0.1 (±0) |
| Aspartate | 418 | 3.8 (±0.3) | 0.5 (±0.2) | 11.3 (±0.3) | 0.5 (±0) |
| Glutamate | 432 | 6.4 (±0.3) | 2.0 (±0.4) | 10.5 (±0.3) | 0.4 (±0) |
| Fatty acid C16:1 | 74 | 2.8 | 1.5 | 11.0 | 2.2 |
| Fatty acid C18:1 | 74 | <1.0 | 0.5 | 5.7 | 4.4 |

Each value depicts the $^{13}$C-excess [%] plus standard deviation ($\sigma$) in the respective amino acids as the mean of three different cultures analyzed three times each with GC-MS. The excess is the increase of $^{13}$C per carbon atom compared to the natural abundance. For the GC-MS measurements of fatty acids, the three separate cultures were combined to increase the amount of substance. Amino acids were measured as TBDMS-derivatives and fatty acids were measured as methyl esters. Hence, values given for the subsequent GC-MS measurements depict the mean from three different samples and standard deviations could not be calculated separately. The fragment [M-57]$^+$ (having lost a tert-butyl moiety from the TBDMS derivatives) was used for the $^{13}$C-excess quantification in case of amino acids where M is the mass of the molecule without any fragmentation, the structures of the fragments can be seen in Fig. S9. For fatty acids, typical fragments with a mass of 74 (McLafferty fragments, see Fig. S8) were measured.

(compound 1 in Fig. 1) as sole electron source. All enzymes could be measured in the oxidative direction indicating a fully functional TCA cycle (Table 1). An exemplary measurement is shown in Fig. S1. All but two enzymes could also be measured in the reductive direction, including the key enzyme 2-oxoglutarate:ferredoxin oxidoreductase which, however, is generally assumed to be fully reversible[27–30]. We could also measure the reverse citrate synthase reaction producing acetyl-CoA from citrate and CoA with an activity of 0.05 ± 0.008 nmol min$^{-1}$ mg$^{-1}$ protein, indicating a completely reversible TCA cycle. Neither a gene for ATP-dependent citrate lyase (ACL) nor for citryl-CoA lyase (CCL) was identified in the genome. One of these enzymes would be a likely candidate for citrate synthesis in a putative rTCA, catalyzing the reverse of the citrate synthase reaction. Their absence does not necessarily exclude a functional rTCA, because there are other options possible, like the roTCA mentioned above. This cycle uses the citrate synthase in the reductive direction, but with much lower enzyme activities than in the oxidative direction.

Generally, the enzyme activities of the TCA cycle were very low in both directions (Table 1), corresponding to the very slow growth rate of N47 with a doubling time of approximately eight days and the equally low rate of naphthalene carboxylase (0.3 nmol min$^{-1}$ mg$^{-1}$ ≙ mU mg$^{-1}$) (see Fig. S2)[11], the first enzyme in the naphthalene degradation pathway. Even though the enzyme activities appear very low, they are still in comparable range to other studied organisms. For example, enzyme activities of 36 nmol min$^{-1}$ mg$^{-1}$ and 10 nmol min$^{-1}$ mg$^{-1}$ were reported for isocitrate dehydrogenase and aconitase, respectively, in cell extracts of *Aquifex pyrofilus* growing on H$_2$, CO$_2$, O$_2$, and thiosulfate[31]. The big range of activities among the enzymes of the TCA cycle seems striking, but this appears to be not uncommon, since enzyme activities of the TCA cycle can range over two orders of magnitude[21,31–33]. This might be caused by assay conditions mimicking the natural conditions for some enzymes better than for others. Furthermore, the low specific activities measured for N47 reflect the extremely slow growth with doubling times of one to two weeks. Nevertheless, the fact that the enzyme activity of 2-oxoglutarate:ferredoxin oxidoreductase was similar to the activities of the other TCA enzymes indicated that the TCA cycle was closed and fully reversible in strain N47.

A functional WLP was demonstrated by measuring the activities of the key enzymes carbon monoxide dehydrogenase and formate dehydrogenase, which showed much higher activities than the TCA cycle enzymes (Table 1). An overview of the TCA cycle and the WLP with the measured enzyme activities can be found in Fig. S3. The 10 to 50-fold higher enzyme activities of the WLP compared to TCA enzyme activities might indicate that the WLP is used for catabolism, i.e., acetyl-CoA oxidation. Nevertheless, it could theoretically be that strain N47 is using the WLP for both acetyl-CoA oxidation and CO$_2$ fixation. However, this would imply that the WLP is in absolute equilibrium which is unlikely due to the fact that the oxidation of acetyl-CoA is coupled to sulfate reduction.

**Naphthalene oxidation and CO$_2$ fixation.** To analyze to which extent the N47 cells utilized the TCA cycle or the WLP for oxidation of acetyl-CoA or possibly for CO$_2$ fixation, N47 was grown with either 10 mol% H$^{13}$CO$_3^-$ and unlabeled (natural $^{13}$C-abundance of approx. 1%) naphthalene or 10 mol% [U-$^{13}$C$_{10}$]naphthalene and bicarbonate buffer (H$^{12}$CO$_3^-$) with natural $^{13}$C-abundance (~1%). Due to the long doubling time of N47 (8 days), the cells were grown for 60 days under these conditions to obtain sufficient cell mass for isotopologue analysis with gas chromatography-mass spectrometry (GC–MS). After 60 days, the bacteria were harvested and a fraction of the pellet was hydrolyzed under acidic conditions to yield amino acids. Another fraction was extracted for fatty acids by treatment with n-hexane (see Materials and Methods in Supplementary Materials).

Amino acids derive from central metabolic intermediates, such as pyruvate, acetyl-CoA, oxaloacetate and 2-oxoglutarate. Since these intermediates are main constituents of the core metabolic network, the labeling patterns of amino acids are qualified to display the patterns of their respective precursors, and, thus, provide information about central metabolic fluxes. For example, aspartate and glutamate reflect the patterns of oxaloacetate and 2-oxoglutarate, respectively, main intermediates of the TCA cycle. To afford sufficient amounts of amino acids, the cells were hydrolyzed and the resulting amino acids, for example from proteins, were silylated into volatile tert-butyldimethylsilyl (TBDMS) derivatives[34]. GC-MS analysis of these derivatives measured in single ion mode (SIM) (see Figs. S4–S7) revealed the abundances of isotopologues containing one or two $^{13}$C-atoms (M+1 and M+2, respectively) more ("excess") than what would be expected in natural abundance material (containing ~1% or 0.01% M+1 or M+2 specimens, respectively) (Table 2). Hence, the natural $^{13}$C abundance is subtracted for all data.

If acetyl-CoA from naphthalene oxidation is incorporated into intermediates of the TCA cycle in experiments with [U-$^{13}$C$_{10}$]naphthalene, these compounds and the amino acids and fatty acids formed from them must reflect the $^{13}$C-isotope ratio of the supplied [U-$^{13}$C$_{10}$]naphthalene, forming substantial amounts of $^{13}$C$_2$-isotopologues (M+2) (see below). Notably, $^{13}$C$_2$-acetyl-CoA would be formed by degradation of the [U-$^{13}$C$_{10}$]naphthalene substrate leading to an overall increase of the molecular mass by two mass units (M+2) in metabolic products using acetyl-CoA as a building block.

Assuming that a mixture of 10 mol% $^{13}$C$_{10}$-naphthalene and 90 mol% unlabeled naphthalene is converted into acetyl-CoA, ca. 10 mol% of the M+2 isotopologue (i.e., $^{13}$C$_2$-acetyl-CoA) would be expected. Consequently, amino acids which are directly formed from acetyl-CoA should display roughly the same $^{13}$C-fractions. However, this was not seen in our data, where the fraction of M+1 was always higher than the M+2 fraction in the $^{13}$C-enriched compounds (Table 2). This suggests that most of the $^{13}$C$_2$-acetyl-CoA that is produced in the catabolism from $^{13}$C$_{10}$-naphthalene oxidation is neither carboxylated into $^{13}$C$_2$-pyruvate serving as the precursor

**Fig. 2 | Expected and observed label distribution in GC-MS measurements of amino- and fatty acids.** The expected **A** and observed **B** results of experiments with $^{13}C$-labeled naphthalene and the expected **C** and observed **D** results of experiments with $^{13}CO_2$. Red lines indicate an expected increase by 2 mass units resulting from degradation of fully labeled naphthalene into acetyl-CoA and the position of the two labeled carbon atoms. A thin red line indicates that the found labeling is not as high as it was expected. Blue dots indicate an increase by 1 mass unit (M+1) and the position of the labeled carbon atom. The columns next to the amino or fatty acids show the measured ratio of M+2 (red) and M+1 (blue). With **1**: naphthalene, **2**: acetyl-CoA, **3**: exemplary fatty acid, **5**: alanine.

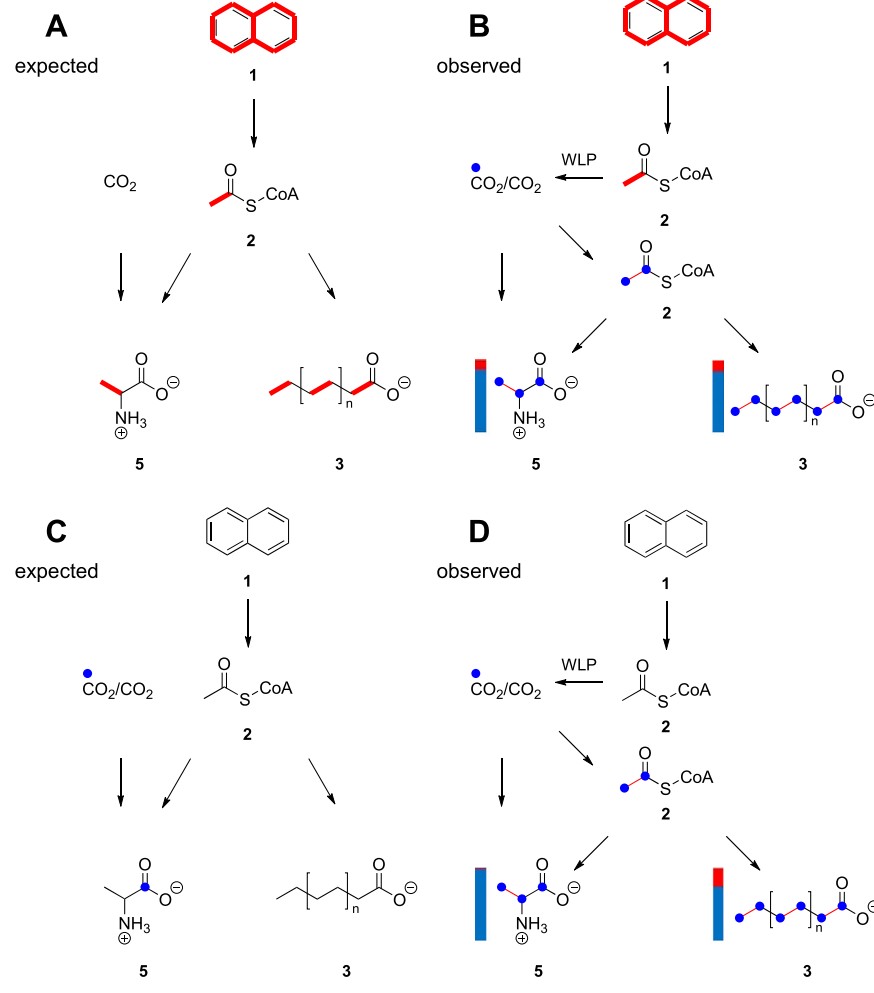

for alanine, nor does it enter the TCA cycle at high rates providing $^{13}C_2$-2-oxoglutarate (serving as precursor for $^{13}C_2$-glutamate) and $^{13}C_2$-oxaloacetate (serving as precursor for $^{13}C_2$-aspartate). Rather, most of the formed $^{13}C_2$-acetyl-CoA seems to be oxidized to $^{13}CO_2$ via the WLP, reducing the M+2 fraction, and increasing the M+1 fraction in the biosynthetic products (Fig. 2). Since 10% of the supplied naphthalene was fully $^{13}C$-labeled, also 10% $^{13}C_2$-labeled acetyl-CoA could be formed from degradation of the aromatic tracer. However, if $^{13}C_{10}$-naphthalene is degraded into $^{13}CO_2$, the unlabeled $CO_2$ in the medium and the gas phase has to be taken into account. Consequently, a maximum of 3.9% $^{13}C$-labeled $CO_2$ could be produced via the WLP (see calculations in Materials and Methods), finally increasing the $^{13}C$-content of the bicarbonate in the medium over time (Fig. S2). Hence, the formation of M+1 labeled acetyl-CoA and its downstream amino acid products in significant amounts (Table 2) provided clear evidence for acetyl-CoA formation via $CO_2$-fixation and naphthalene oxidation primarily to $CO_2$ rather than the use of acetyl-CoA directly from naphthalene for anabolism (see also Fig. 1).

**Experiment with $^{13}CO_2$.** This hypothesis was confirmed by labeling experiments using $H^{13}CO_3^-$ as a tracer. When the bicarbonate buffer was labeled with 10 mol% $^{13}C$ and the substrate naphthalene had natural $^{13}C$-abundance, the overall $^{13}C$-excess values of amino acids (Table 2) corresponded well to the expected maximum value for $^{13}C$-content of the total inorganic carbon in the culture bottle if naphthalene was completely degraded to $CO_2/HCO_3^-$ (5.1%) (see calculation in Materials and Methods). If an amino acid has three carbon atoms like alanine, and the probability that a carbon atom is $^{13}C$-labeled is 5.1%, the statistical probability of a M+1

labeled alanine being formed would be 13.8% (see Material and Methods). Due to the very slow growth of N47 it is very likely that there are still amino acids present in proteins that were formed during growth on unlabeled substrate in the pre-culture decreasing the observed values. In addition, the pretreatment of the medium using unlabeled $CO_2$[11] could also have decreased the $^{13}C$ ratios. Nevertheless, the presence of mostly M+1 labeled products (Table 2) indicates that most of the acetyl-CoA, which is used for anabolism and synthesis of amino acids from pyruvate or intermediates of the TCA cycle, comes from $CO_2$ fixation rather than from naphthalene degradation, which agrees well with the data observed in the experiment with $^{13}C$-labeled naphthalene (see above). Hence, both labeling experiments confirmed that naphthalene oxidation proceeds through acetyl-CoA which is fueled into the WLP for $CO_2$ formation (roughly three orders of magnitude higher enzyme activity according to Table 1).

The labeling patterns of fatty acids must reveal the isotope profile of their basic precursor, acetyl-CoA. Therefore, we also analyzed the $^{13}C$-labeling patterns in fatty acids that were extracted from the cells by hexane treatment. In the GC chromatogram of the fatty acid methyl esters, C16:1 and C18:1 were well separated and further analyzed. The $^{13}C$-accumulation was measured from corresponding McLafferty fragments in the mass traces with m/z values = 74 (Fig. S8). This fragment contains C1 and C2 of the original fatty acid and is therefore equivalent to the acetyl moiety in acetyl-CoA.

The $^{13}C$-composition was characterized by high M+1 fractions in both labeling experiments (Table 2). This finding provided further evidence that acetyl-CoA used for biosynthesis was mainly formed from $CO_2$. The detected minor M+2 fractions in fatty acids from the labeling experiment

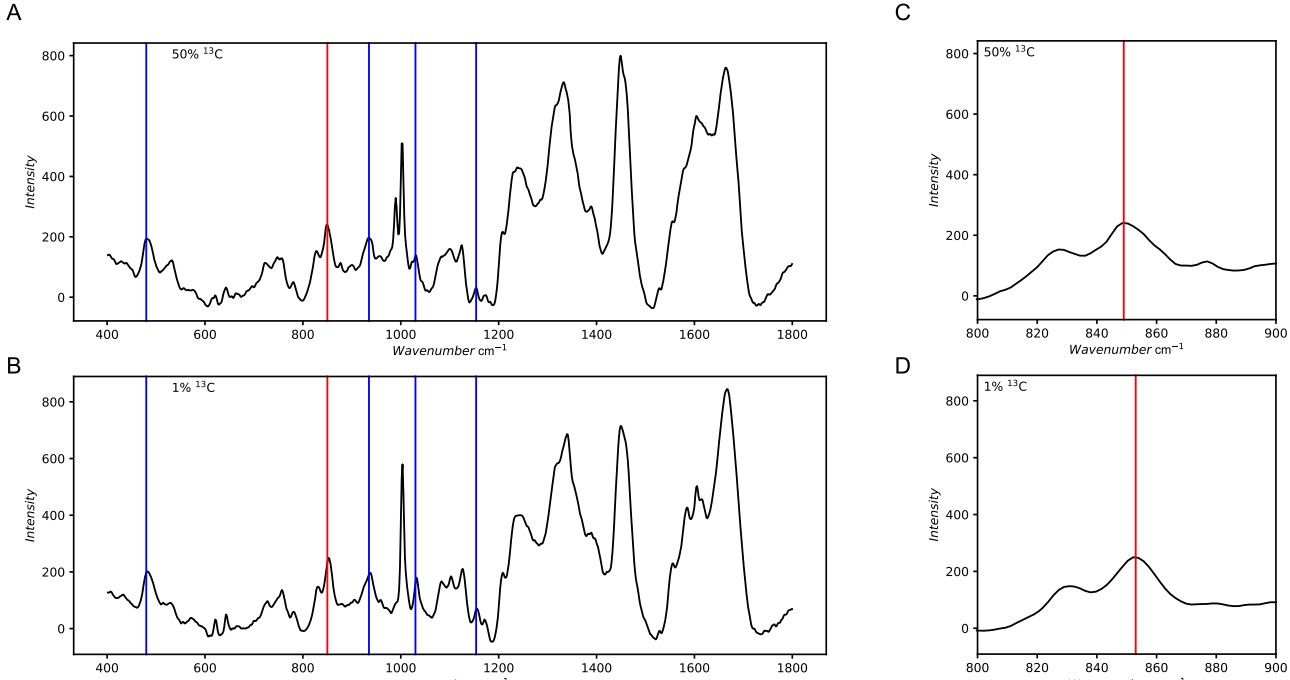

**Fig. 3 | Raman spectra and enlargement of the glycogen associated bands of culture N47 grown with ¹³C enrichment or only natural abundance of ¹³C.** Raman spectra of culture N47 grown with either **A** 30 mM ¹³C-labeled bicarbonate (50 atom % ¹³C) or **B** non-labeled bicarbonate (1% ¹³C, natural abundance). **C**, enlargement of the glycogen-associated Raman band of cells grown with ¹³C-labeled bicarbonate (50 atom%) or **D**, non-labeled bicarbonate. Blue lines mark the characteristic bands representing polyphosphate, while the band marked in red is a prominent characteristic of glycogen. The presented spectra depict means of 20 spectra measured from different cells each.

with ¹³C-naphthalene could reflect that, in fatty acid metabolism, the separation of anabolism (usage of acetyl-CoA from $CO_2$ fixation) and catabolism (usage of acetyl-CoA from $^{13}C_{10}$-naphthalene) is not absolute. If the acetyl-CoA is built from $CO_2$ with 10 atom% ¹³C label, the probability of the M+2 fraction with two labeled carbon atoms is about 1% through statistical coupling.

*Potential other influences:. Isotope exchange reactions:* For the interpretation of the labeling data, an exchange reaction of $C_1$ (carboxyl C) in $^{13}C_2$-acetyl-CoA via the WLP has to be considered as well. Such an exchange reaction of the carboxyl group of acetyl-CoA can be catalyzed by carbon monoxide dehydrogenase and would apply to the whole acetyl-CoA pool in the cell. If the exchange reaction was highly efficient, no significant M+2 fractions in amino acids and fatty acids should be observed in the experiment with $[U-^{13}C_{10}]$naphthalene because the carboxyl group of acetyl-CoA could exchange whereas the methyl group could not. A total isotope exchange is very unlikely because the WLP is pulled toward oxidation by sulfate reduction. Nevertheless, if isotope exchange would occur, the label incorporation into amino acids from $[U-^{13}C_{10}]$naphthalene should account for 50% of the label from $^{13}C_2$-acetyl-CoA. Since 10% of the naphthalene was labeled, the amino acids should still show a label incorporation of 10% M+1 label from the methyl group of acetyl-CoA plus the label incorporation from the bicarbonate buffer, which was clearly not the case (Table 2). In the case of 10% ¹³C-labeled bicarbonate buffer, the label incorporation should only be less than 10% because still the carbon from the methyl group of acetyl-CoA should be non-labeled. This also did not agree with the obtained isotope incorporation in Table 2.

*Influence of naphthalene carboxylation:* In a scenario where acetyl-CoA is made from naphthalene, one has to consider that the degradation pathway involves the carboxylation of naphthalene ($C_{10}$ compound) to naphthoic acid ($C_{11}$ compound), as the initial activation step. Therefore, the carboxyl group could be labeled to a maximum of 10% in experiments with 10% $^{13}CO_2$. Thus, only 10 mol% of one carbon atom in the resulting naphthoic acid could be labeled from $^{13}CO_2$ and the theoretical label incorporation into

naphthalene-derived acetyl-CoA is only less than one percent $\left(\frac{1}{11} \cdot 10\% = 0.9\%\right)$, which is significantly smaller than the observed values for the acetyl-CoA precursor in fatty acids (6 – 11% M+1) (Table 2). Therefore, the ¹³C-data observed in the experiment with $^{13}CO_2$ cannot be explained by the initial carboxylation reaction of naphthalene only.

Taken together, the labeling patterns in amino acids and fatty acids indicated that strain N47 mainly oxidizes naphthalene to $CO_2$ via acetyl-CoA and the WLP, and synthesizes acetyl-CoA for anabolism de novo from $CO_2$ via a reversed TCA. Hence, it indicates a chemoorganoautotrophic metabolism.

## Storage compounds in N47

Separating anabolism and catabolism in such a way would require a possibility to store the energy acquired from naphthalene degradation until it is needed for anabolic processes, for example in storage compounds like polyphosphates or glycogen. To investigate the presence of storage compounds, Raman spectroscopy of single cells was applied to N47 cultures that were grown with bicarbonate buffer labeled with 50 atom% ¹³C or at natural ¹³C abundance. There was no indication for polyhydroxybutyrate (PHB) which has a main characteristic band at 1720 cm⁻¹ due to C = O stretching vibrations[35,36] (Fig. 3). Peaks at 480 cm⁻¹ (bending vibrations of phosphate groups) and 700–800 cm⁻¹ (stretching vibrations of P-O-P bonds) indicated the presence of polyphosphate. Other typical bands at 935 cm⁻¹ (symmetric stretching vibrations of phosphate groups in the polymer chain), in the region of 1000–1050 cm⁻¹ (symmetric stretching vibrations of P-O bonds) and at 1154 cm⁻¹ (asymmetric stretching vibrations of P-O-P) were not pronounced indicating that polyphosphate is either not a true constituent or only present in small amounts[37–39]. However, the presence of glycogen was confirmed by a band that typically appears around 850 cm⁻¹, which is assigned to vibrations of glycosidic bonds[40]. When N47 was grown with ¹³C-labeled bicarbonate (50 atom%, Fig. 3A, C) or with natural abundance of ¹³C (Fig. 3B, D), a red-shift was observed in the glycogen-associated Raman band from 854 cm⁻¹ to 849 cm⁻¹, which indicates that the storage compound glycogen is built

**Table 3 | Organisms with genomes encoding for both a complete TCA cycle (KEGG module: M00009) and WLP (KEGG module: M00377) as well as the gene for 2-oxoglutarate:ferredoxin oxidoreductase [EC:1.2.7.3 1.2.7.11]**

| Organism | Taxonomy | Environment | Genes for HC degradation | BioProject |
|---|---|---|---|---|
| *Candidatus* Desulfobacterium strain N47[10,12,14,22,48] | Bacteria; Desulfobacterota; *Desulfobacteria; Desulfobacterales; Desulfosarcinaceae* | Enriched from contaminated aquifer near Stuttgart, Germany | yes | FR695864-FR695880 & GU080088, GU080089, GU080116-GU080137, GU080090-GU080115 PRJNA1066605 |
| TRIP_1[41] | Bacteria; Desulfobacterota; Desulfobacteria, *Desulfobacterales; Desulfobacteraceae* | Pitch Lake in Trinidad-Tobago | yes | PRJNA479783 |
| Strain *NaphS2* DSM 14454[13,42] | Bacteria; Desulfobacterota | Black anoxic marine sediment | yes | PRJNA46635 |
| *Desulfosarcina alkanivorans* PL12[44] | Bacteria; Thermodesulfobacteriota; Desulfobacteria; *Desulfobacterales; Desulfosarcinaceae; Desulfosarcina* | Obtained from oil-polluted marine sediments of Shuaiba, Kuwait | yes | PRJDB8884 |
| *Desulfatibacillum aliphaticivorans* AK-01 (*Desulfatibacillum alkenivorans* AK-01)[45] | Bacteria; Thermodesulfobacteriota; Desulfobacteria; *Desulfobacterales; Desulfatibacillaceae; Desulfatibacillum* | Petroleum-contaminated sediment collected from the Arthur Kill, an intertidal waterway between Staten Island and New Jersey | yes | PRJNA19449 |
| *Desulfococcus oleovorans* Hxd3[46] | Bacteria; Thermodesulfobacteriota; Desulfobacteria; *Desulfobacterales; Desulfosudaceae; Desulfosudis* | Oil/water mixture from an oil production plant | yes | PRJNA18007 |
| *Desulfoluna limicola* ASN36[75] | Bacteria; Thermodesulfobacteriota; Desulfobacteria; *Desulfobacterales; Desulfolunaceae; Desulfoluna* | Isolated from a sample of sediment collected in a brackish lake, Lake Akkeshi in Japan (43.05 deg N 144.89 deg E). | potentially | PRJDB11192 |
| *Desulfomonile tiedjei* DSM 6799[76] | Bacteria; Thermodesulfobacteriota; Desulfomonilia; *Desulfomonilales; Desulfomonilaceae; Desulfomonile* | Sewage sludge | potentially | PRJNA43133 |
| *Candidatus* Brocadia pituitae 317325-1[77] | Bacteria; Planctomycetota; *Candidatus* Brocadiia; *Candidatus* Brocadiales; *Candidatus* Brocadiaceae; *Candidatus* Brocadia | Obtained from the anammox bacterial community (ABC) metagenome | no | PRJDB4337 |
| *Candidatus* Jettenia sp. AM49[77] | Bacteria; Planctomycetota; *Candidatus* Brocadiia; *Candidatus* Brocadiales; *Candidatus* Brocadiaceae; *Candidatus* Jettenia | Observed at the walls of the formate reactors | no | PRJNA556270 |
| SEEP-SRB1[43] | Bacteria; Desulfobacterota; Desulfobacteria, *Desulfobacterales; Desulfosarcinaceae* | Marine sediments | potentially | PRJNA326769 PRJNA290197 |

Organisms were found by KEGG database brite hierarchy search[73] and literature hits. Genomes can be accessed via BioProject numbers. Last KEGG database search: 17.07.2023. HC, hydrocarbon degradation. In column "Genes for HC degradation" the term "potentially" indicates that single genes involved in hydrocarbon degradation pathways were found in the genome or that close related organisms are capable of hydrocarbon degradation, but oil degradation was not found to be approved by experiments for this specific organism.

from $CO_2$. The phenylalanine Raman band at 1004 cm$^{-1}$ for unlabeled phenylalanine showed a red-shift to lower wavenumbers upon incorporation of $^{13}C$, due to the mass effect of the heavier isotope. This shift further qualitatively supports the $^{13}C$-incorporation from $^{13}CO_2$ into amino acids built from TCA intermediates.

## Reverse TCA cycle and WLP in other microorganisms

Assuming that the extraordinary feature of possessing a complete roTCA cycle and WLP is a generic genetic prerequisite for the new chemoorganoautotrophic metabolism of complex aromatic substrates, we performed a KEGG database and literature screening of genomes to find if this metabolism occurs more frequently and is not unique to strain N47.

In total, we found eleven hits where both a complete TCA cycle containing 2-oxoglutarate:ferredoxin oxidoreductase and WLP are encoded in the genomes or were even expressed in the proteome (Table 4 including the supporting references). KEGG database search resulted in seven genomes encoding both pathways. Five of them were classified as Thermodesulfobacteriota and two as Planctomycetota. By literature research, we found the metagenomes and proteomes of three more organisms, the phenanthrene-degrading *Candidatus* Desulfatiglans TRIP (Pseudomonadota)[41] and the naphthalene-degrading strain NaphS2 (Pseudomonadota) (DSM 14454)[42], which are related to *Candidatus* N47, and the proposed metabolism of *SEEP-SRB1* based on metagenomic analysis[43], the eleventh is N47, presented in this study (Table 3). All of these eleven bacterial genomes were extracted from anaerobic microorganisms

encoding genes for either dissimilatory sulfate reduction (six Thermodesulfobacteriota and *SEEP_SRB1* genomes) or anaerobic ammonium oxidation (ANAMMOX, the two candidates *Brocadia pituitae* and *Jettenia* sp. AM49) (Table 4). At least five genomes originated from oil-related or contaminated environments. Six organisms are capable of hydrocarbon degradation, using substrates such as n-hexane (*Desulfosarcina alkanivorans* strain PL12)[44], n-alkane, 1-pentadecene and 1-hexadecene (*Desulfatibacillum aliphaticivorans* strain AK-01)[45], C12-C20 alkanes (*Desulfococcus oleovorans* Hxd3)[46,47], phenanthrene (TRIP_1)[41], and naphthalene (strain NaphS2, *Cand.* strain N47)[10,12–14,22,42,48], proven by cultivation experiments.

Although the special feature of having both a complete reversible TCA cycle and WLP is present in sulfate-reducing microorganisms oxidizing alkanes, naphthalene, or phenanthrene this does not seem to be linked exclusively to anaerobic hydrocarbon degraders. The other five genomes encoding a putative reversible TCA cycle and WLP belong to anaerobic ammonium oxidizers (ANAMMOX) (*Candidatus* Brocadia pituitae or *Candidatus* Jettenia) or syntrophic partners of archaea that perform anaerobic oxidation of methane (ANME) (Table 4)[43]. These genomes contain only single genes possibly related to degradation of hydrocarbons but no complete anaerobic hydrocarbon degradation pathway. Hydrocarbon degradation was also not proven in cultivation experiments of these strains, so far.

The analysis of the presented genomes suggests that the exceptional feature of possessing both a functional reversible TCA cycle and a WLP is a

feature of sulfate-reducing, hydrocarbon-degrading microorganisms and some others such as ANAMMOX. Although chemoorganoautotrophic degradation of polycyclic aromatic substrates has only been shown for strain N47 in this work, having the two central pathways might be correlated to this new metabolism. Nevertheless, the genomic search here is not comprehensive, and other microorganisms might exist that show the same genetic potential.

## Separating anabolism and catabolism

The observation of chemoorganoautotrophic growth raises the question of how this metabolism can work and what are possible benefits. If the oxidation of acetyl-CoA to $CO_2$ and the fixation of $CO_2$ to acetyl-CoA would be two completely separate processes, they would require a spatial or a temporal separation of the two acetyl-CoA pools to prevent futile cycles. An example for spatial separation exists in eucaryotes where pyruvate from glycolysis is transported from the cytosol into mitochondria and cleaved into acetyl-CoA and $CO_2$. Acetyl-CoA then enters the TCA cycle for total oxidation. When the energy status of the cells is high, citrate is exported from the mitochondria into the cytosol where acetyl-CoA is released by ATP citrate lyase and can be used for example for fatty acid synthesis. Hence, there exists an energy level-dependent regulation of acetyl-CoA metabolism for catabolism and anabolism in two different compartments of the eukaryotic cell[49]. However, if the chemoorganoautotrophic N47 cells would use a similar spatial separation they would need defined compartments for one of the acetyl-CoA pools similar to, for example carboxysomes[50]. So far, we were not able to visualize any defined compartments inside the N47 cells, nor could we identify genes that are commonly associated with microbial carboxysomes (carboxysome shell (csoS) genes like *CsoS2, ccmK, ccmL* or the Rubisco enzyme)[51,52].

An example for temporal separation of anabolism and catabolism are yeasts[53]. Carbon source-limited chemostat cultures of *Saccharomyces cerevisiae* exhibit the yeast metabolic cycle where the oxidation of glucose (respiratory phase) is strictly separated in time from the cell division phase where no respiration is going on. This metabolic cycle also includes the mitochondria which only proliferate during the non-respiratory phase. Such a separation in time would also be possible in strain N47 but would require a thorough regulation, similar to yeast[53].

This leads to the question why N47 and potential other microorganisms use the chemoorganoautotrophic metabolism discovered here. It is conspicuous that sulfate-reducing hydrocarbon degraders, and all other microorganisms identified in the genome search containing both a reversible TCA cycle and WLP, are extremely slow-growing microorganisms, which can be linked to very low substrate availability or thermodynamic energy gain in the environment. A speculative scenario explaining the observations would be that these microorganisms cannot simultaneously metabolize and divide, like fast-growing microorganisms in a batch culture, because they conserve very little energy per time. Caused by a lack of energy, they could get stuck in an energy-demanding cell division cycle without a chance to complete this cycle, which might be deadly. Consequently, the cells might need to accumulate energy storage compounds (for example polyphosphate or glycogen) over long time spans until they can divide. Glycogen could be produced by known pathways from the acetyl-CoA produced in the roTCA. In the extremely oligotrophic environments inhabited by these organisms, there might not be enough carbon source readily available, forcing the cells to fix abundant carbon from $CO_2$ for growth. Such a time shift between catabolism and autotrophic anabolism similar to the yeast metabolic cycle[53] would match the $^{13}C$-labeling patterns observed in our experiments. The observation of glycogen as a storage compound in N47 that is created from $CO_2$ supports this speculative hypothesis. Although the cells would "waste" some of their energy for $CO_2$ fixation, the time shift of catabolism and anabolism might be a safer strategy in extremely oligotrophic environments.

Oxidation of one molecule of naphthalene theoretically allows for conservation of 2–3 ATP[4,54]. Since growth yields of such anaerobic

organisms are typically low (10 g biomass/mol ATP)[4], between 0.15 to 0.23 g of biomass (~0.07–0.11 g carbon) can be built per gram naphthalene, resulting in a theoretical yield coefficient for carbon of roughly 10%. For $CO_2$ fixation, which likely happens via the roTCA, further energy has to be invested (1 ATP per $CO_2$[21]) reducing the yield coefficient even more, but this might be justifiable as the safer strategy for the cells. Nevertheless, only a minor portion of the $CO_2$ produced by naphthalene oxidation is needed for biomass production via $CO_2$ fixation.

The stable isotope-labeling experiments with $^{13}CO_2$ or $^{13}C$-naphthalene revealed that acetyl-CoA produced from naphthalene degradation was incorporated only to a minor extend as an intact $C_2$-unit into amino acids derived from intermediates of the TCA cycle, or into fatty acids. In contrast, a strong $^{13}C$-incorporation into the same amino acids and fatty acids was observed when $^{13}CO_2$ was added to the cultures. The $^{13}C$ labeling patterns of amino acids and fatty acids indicated the use of the WLP for oxidation of acetyl-CoA that is derived from naphthalene oxidation. As the pathway cannot run in both directions simultaneously, it is very likely that the $CO_2$ fixation happens via the roTCA. Hence, *Candidatus* Desulfobacterium strain N47 points towards a new type of metabolism where it oxidizes the primary electron and carbon source naphthalene mainly to $CO_2$ via the WLP while using a reverse TCA cycle to fix $CO_2$ for biosynthesis of acetyl-CoA and its downstream products. Therefore, the data point to the likelihood of N47 being the first example for chemoorganoautotrophic growth with complex organic substrates, in this case polycyclic aromatic hydrocarbons.

## Methods

### Spectrophotometric activity assays

Enrichment culture N47 was grown and harvested as described previously and cell free extracts (cfe) were prepared by french press[11]. Cultures were started with 10% of the culture volume as inoculum from precultures. Due to the extremely slow growth, cells were continuously cultivated in large numbers of two liter bottles and the needed volume of culture was freshly harvested.

Cell-free extracts (cfe) were prepared fresh from freshly harvested cells at the beginning of each experiment. Two to three 1.6 L cultures were needed per experiment to produce enough cfe for one assay with all necessary controls.

All enzyme assays were carried out spectrophotometrically (Cary 50 Bio, Varian, Darmstadt, Germany) in anoxic, sealed quartz glass cuvettes (Carl Roth GmbH, Karlsruhe, Germany) under an atmosphere of $N_2$ at 30 °C. The assay mixture was prepared in a glovebox (MBraun, Garching, Germany) directly in the cuvettes and the substrate used to start the reaction was injected with a syringe through the rubber stopper directly before measuring. All assays were carried out in triplicates with controls lacking one of the substrates. Assay compositions were adapted and modified from previous publications and are listed below for clarity[21,25,31,32,55,56]. The raw data of the aconitase and isocitrate dehydrogenase measurement is shown in Fig. S1 as an example.

Citrate synthase (oxidative direction): 1.5 mM oxaloacetate (omitted in the controls), 0.1 mM acetyl-CoA, 0.1 mM DTNB, 0.1 mM AMP, 1.5 mM $Na_2HPO_4$, 3 mM $MgCl_2$, 100 mM TRIS; 18% (v/v) cfe – ~1.0 mg/l total protein; pH 7.2; measured at 412 nm (CoA-SH)[32]

Activity was calculated after 60 min

Aconitase + isocitrate dehydrogenase (oxidative direction): 5 mM $NADP^+$, 1 mM citrate (omitted in the controls), 100 mM MOPS, 15 mM $MgCl_2$; 20% (v/v) cell free extract (cfe) – ~1.0 mg/l total protein; pH 7.3; measured at 340 nm (NADPH)[31]

Activity was calculated after 1 min

2-oxoglutarate:ferredoxin oxidoreductase (oxidative direction): 2.5 mM 2-oxoglutarate (omitted in the controls), 0.5 mM CoA-SH, 1 mM methyl viologen, 0.025 mM $Na_2S_2O_4$, 100 mM MOPS, 10 mM $MgCl_2$, 5 mM DTT; 20% (v/v) cfe – ~0.6 mg/l total protein; pH 7.0; measured at 600 nm (reduced methyl viologen)[25]

Activity was calculated after 1 min

Succinate dehydrogenase (oxidative direction): 1 mM succinate, 0.1 mM menadione (omitted in the controls), 100 mM TRIS, 1.5 mM $MgCl_2$; 20% (v/v) cfe – ~0.9 mg/l total protein; pH 7.2; measured at 250 nm (fumarate)[32]

Activity was calculated after 9 min

Fumarase (combined with malate dehydrogenase; oxidative direction): 2 mM fumarate (omitted in the controls), 2 mM $NAD^+$, 100 mM TRIS, 1.5 mM $MgCl_2$; 20% (v/v) cfe – ~0.6 mg/l total protein; pH 7.2; measured at 340 nm (NADH)[56]

Activity was calculated after 30 min

Malate dehydrogenase (oxidative direction): 5 mM malate (omitted in the controls), 2 mM $NAD^+$, 100 mM MOPS, 10 mM $MgCl_2$, 5 mM DTT; 10% (v/v) cfe – ~0.9 mg/l total protein; pH 7.0; measured at 340 nm (NADH)[32]

Activity was calculated after 5 min

Isocitrate dehydrogenase (reductive direction): 1 mM 2-oxoglutarate (omitted in the controls), 0.4 mM NADPH, 100 mM TRIS, 1.5 mM $MgCl_2$; 10% (v/v) cfe – ~1.2 mg/l total protein; pH 7.2; measured at 340 nm (NADPH)[31]

Activity was calculated after 1 min

Succinate dehydrogenase (reductive direction): 19.23 mM fumarate (omitted in the controls), 0.38 mM methyl viologen (previously reduced with DTT), 50 mM TRIS; 20% (v/v) cfe – ~1.2 mg/l total protein; pH 7.4; measured at 600 nm (reduced methyl viologen)[56]

Activity was calculated after 0.7 min

Fumarase (reductive direction): 2 mM malate (omitted in the controls), 100 mM TRIS, 1.5 mM $MgCl_2$; 20% (v/v) cfe – ~1.1 mg/l total protein; pH 7.2; measured at 250 nm (fumarate)

Activity was calculated after 60 min

Malate dehydrogenase (reductive direction): 0.2 mM 2-oxoglutarate, 0.4 mM NADH, 100 mM MOPS, 10 mM $MgCl_2$, 5 mM DTT; 23% (v/v) cfe – ~0.9 mg/l total protein; pH 7.0; measured at 340 nm (NADH)[56]

Activity was calculated after 5 min

CO dehydrogenase (WLP): 0.4 mM methyl viologen, 0.6 mL CO added to the headspace (omitted in the controls and replaced by $N_2$), 100 mM TRIS; 20% (v/v) – ~0.6 mg/l total protein; pH 10.0; measured at 600 nm (reduced methyl viologen)[55]

Activity was calculated after 0.5 min

Formate dehydrogenase (WLP): 3 mM formate (omitted in the controls), 0.4 mM methyl viologen, 2 mM DTT, 1 mM $MgCl_2$, 100 mM TRIS; 20% (v/v) cfe) – ~1.5 mg/l total protein; pH 8; measured at 600 nm (reduced methyl viologen)[25]

Activity was calculated after 0.5 min

In the oxidative direction of the TCA cycle, the activities of aconitase and isocitrate dehydrogenase were measured in a combined assay of the two reactions, and the fumarase activity was measured in combination with malate dehydrogenase, while malate dehydrogenase could also be measured in a single reaction without fumarase. 2-oxoglutarate:ferredoxin oxidoreductase was not measured in the reverse direction of the TCA cycle, but it is generally assumed that the enzyme is fully reversible.

Discontinuous assays of TCA cycle enzymes were measured with LC-MS analysis of acetyl-CoA. Enzyme assays were performed with 100 mM MOPS/KOH buffer, pH 7.3 with 15 mM $MgCl_2$ and started by addition of 30% (v/v) N47 cell free extract resulting in a final protein concentration of 1.75 mg $mL^{-1}$. The following enzymes were measured:

ATP-citrate lyase: 10 mM citrate, 5 mM ATP, 1 mM CoA-SH; 5 mM NADH or 5 mM $NAD^+$ was added whenever applied.

Reverse reaction of citrate synthase: 10 mM citrate, 1 mM CoA-SH; 5 mM NADH or 5 mM $NAD^+$ was added whenever applied.

Protein concentrations of the cell free extracts were determined according to the Bradford method using a commercially available solution (ROTI®Quant, Carl Roth GmbH, Karlsruhe, Germany)[57].

**LC-MS analysis of acetyl-CoA in discontinuous enzyme assays.** 50 μL aliquots were taken at different time points and the reaction was immediately stopped by addition of a double volume of methanol. Samples were then centrifuged for 30 min at $18,200 \times g$ and 4 °C in an Eppendorf tube before the supernatant was analyzed by LC-MS (LC 2040 C 3D, LCMS2020; Shimadzu, Duisburg, Germany).

LC-MS measurements were performed with 0.1% (w/v) ammonium formate buffer as eluent A and acetonitrile as eluent B. Acetonitrile increased from 5% up to 35% over 15 min at a flow rate of 0.4 mL $min^{-1}$. Samples were separated via a Nucleodur C18 Gravity-SB column (100 × 3 mm, 5 μm particle size, Macherey-Nagel, Düren, Germany) at 35 °C. Nebulizing gas flow in the MS was 1.5 L/min, drying gas flow 12 L $min^{-1}$. Mass spectrometric analysis was carried out with an ESI system and in single ion mode for the expected masses for the target compounds. For measurements of acetyl-CoA, an external calibration was prepared from commercially available substance (PanReac AppliChem, Darmstadt, Germany).

### $^{13}C$ labeling experiments

Culture N47 was grown in a minimal medium in a culture volume of 150 mL in 200 mL serum flasks as described previously[11], containing 0.03% (w/v) naphthalene, which was dissolved in 2,2,4,4,6,8,8-heptamethylnonane (HMN) because of its low solubility in water (~250 μM[58]), and 30 mM $HCO_3^-$. Measuring naphthalene concentrations during the experiments is possible[59,60], but requires withdrawal of further aliquots of the water and HMN phases for analysis, which was avoided here in order to not alter the mass balance in the culture bottles. Hence, we measured the $CO_2$-evolution which directly monitors the amount of naphthalene oxidized to $CO_2$.

In setting 1, naphthalene was a mixture of 10 mol% fully labeled $[U-^{13}C_{10}]$naphthalene (Cambridge Isotope Laboratories, USA), and 90 mol % unlabeled naphthalene (containing the natural $^{13}C$ abundance of 1.1%) whereas $HCO_3^-$ was unlabeled (natural abundance). In setting 2, 10 mol% of $HCO_3^-$ were $^{13}C$-labeled (Cambridge Isotope Laboratories, USA), 90 mol % were unlabeled, whereas naphthalene had the natural abundance of 1.1% $^{13}C$. Isotope ratios in the cultures were always calculated based on total $CO_2$ in the culture bottle, including the headspace, to take the potential equilibration between the headspace and the aqueous phase into account. Additionally, one set of cultures was grown under the same conditions with only natural abundance substrates, to be measured later by GC-MS together with the other cultures and be used in the calculations to correct for the influence of naturally occurring $^{13}C$.

10% $^{13}C$ label was used in both types of labeling instead of 100% because of the principle of the stable isotope ratio measurements of $CO_2$ which would not be possible with 100% $^{13}C$ in the buffer or the gas phase because the precision of the measurements decreases with higher labeling ratios and is optimal at 10% label. Additionally, undesirable statistical couplings are higher in a 100% $^{13}C$ experimental setup. Furthermore, $^{13}C$-labeled naphthalene is very expensive and these costs were reduced by using 10% labeled naphthalene which, however, does not change the sensitivity of the applied methods. Moreover, the same percentage of $^{13}C$-labeled substrate was used in both experiments to make the two experiments comparable.

Every week, the cell number was monitored using a flow cytometer (NovoCyte, Agilent, Santa Clara, USA) and the $CO_2$ evolution was measured using a Delta Ray $CO_2$ Isotope Ratio Infrared Spectrometer (Thermo Fisher Scientific, MA, U.S.A.) with Universal Reference Interface Connect for measuring carbon isotope compositions of $CO_2$ to follow the growth of the cultures. In both settings, naphthalene and $HCO_3^-$ served as the only carbon sources and each was provided at concentrations, which deliver similar amounts of carbon atoms in both settings, i.e., 2.3 mM naphthalene (theoretical concentration if all naphthalene were dissolved in water) and 30 mM bicarbonate. Therefore, these conditions were appropriate to distinguish between the relative contributions of catabolic and anabolic pathways, starting from naphthalene or $CO_2$ (with $CO_2$ being in equilibrium with the pre-offered $HCO_3^-$), respectively.

All samples were taken anoxically with $N_2$ flushed syringes (5.0 grade; sterile Hungate needle, $CO_2$-free). For $CO_2$ development analysis, 12 mL Labco Exertainer glass vials (Labco Limited, U.K.) were pretreated with

100 µL of 85% phosphoric acid, closed with screw caps with butyl rubber septa, and flushed with $CO_2$-free synthetic air (6.0 grade; Air Liquide, Germany). 500 µL aqueous sample were taken with syringes through the butyl stoppers of the inoculated serum flasks and directly transferred into prepared glass vials. $CO_2$ in synthetic air at 414.2 ppm (Air Liquide, Germany) was used for $CO_2$ concentration calibration of the device. $CO_2$ reference gases used for calibration of carbon isotope ratios had $\delta^{13}C$ values of −9.7‰ (Thermo Fisher, Germany) and $x(^{13}C) = 10\%$ (Sigma-Aldrich, Germany). Pure $CO_2$ gas with $x(^{13}C) = 10\%$ was used as working reference gas. The $CO_2$ concentration for reference and sample gas entering the analyzer was set to 380 ppm for optimal precision and $CO_2$-free synthetic air was applied as carrier gas. Each sample was measured for 5 min and the obtained $\delta^{13}C$ values were averaged. The stable carbon isotope data were received as delta values and converted into isotope-amount fraction. For flow cytometry a total of 500 µL of sample was taken from each culture in a sterile, anoxic manner, diluted 1:1000 and measured directly via flow cytometry in technical triplicates. N47 cells were stained for 30 min in the dark with Syto9 at a final concentration of 5 µM. Cell counts were performed using a NovoCyte® Flow Cytometer (equipped with a FITC laser at 488 nm, ACEA Biosciences, Inc.) either by analyzing a fixed sample volume of 50 µL or by counting 12,000 events. Cell counts were counted based on the green fluorescence signal where the threshold was set to 1000.The instrument was calibrated in advance with NovoCyte QC Particles (Agilent Santa Clara, USA). Gating was performed to distinguish cells from background signals, an example is shown in Fig. S10. After 60 days, the cultures were harvested and the cell pellets were freeze-dried for further analysis of amino acids and fatty acids.

To elucidate the carbon metabolic pathways and to investigate the contributions of the respective carbon sources, that is carbon directly derived from degradation of naphthalene and carbon originating from $CO_2$ fixation, in the metabolism of N47, we analyzed the stable isotope incorporation from each of the labeled precursors into the same set of metabolites, that is protein-derived amino acids and fatty acids.

## Calculated $^{13}C$ abundances in the experiments

The maximum content of $^{13}C$ in experiments with $^{13}CO_2$ or [U-$^{13}C_{10}$] naphthalene was calculated using

$$\frac{^{13}C}{^{total}C(naphthalene) + {}^{total}C(HCO_3^-) + {}^{total}C(gasphase)}$$

In a 150 mL bottle, the total amount of $HCO_3^-$ is 0.45 mmol, the total amount of $CO_2$ produced from the degeneration of naphthalene is 3.45 mmol and the amount of $CO_2$ in the gas phase is 0.8 mmol. In both experiments, the proportion of $^{13}C$-labeled substances is 10 mol%. Therefore, the maximal amount of $^{13}C$ in both experiments calculates as follows:

$$^{13}C(CO_2) = \frac{0.45\ mmol}{3.45\ mmol + 4.5\ mmol + 0.8\ mmol} = 5.1\ \%$$

$$^{13}C(naphthalene) = \frac{0.345\ mmol}{3.45\ mmol + 4.5\ mmol + 0.8\ mmol} = 3.9\ \%$$

Pretreatment of the medium with $N_2/CO_2$ and a possible incomplete degradation of naphthalene was not considered in this calculation as these effects were not determined.

## Amino acid preparation for GC-MS

For analysis of protein-bound amino acids, 2 mg of the bacterial sample (freeze-dried cell pellet) were suspended in 1 mL of 6 M hydrochloric acid and hydrolyzed for 15 h at 105 °C. The reaction mixture was dried under a stream of nitrogen at 70 °C and the residue was suspended in 200 µL of 50% aqueous acetic acid using an ultra-sonic bath for 3 min. The solution was applied onto a small column of Dowex 50WX8 (7 × 10 mm; 200–400 mesh, 34–74 µm, $H^+$- form, Alfa Aesar). The column was first washed with 2 mL of

$H_2O$, then eluted with 1 mL of 4 M aqueous ammonia. The ammonia eluate was dried under a stream of nitrogen at 70 °C. The residue was treated with 50 µL of N-(tert-butyldimethylsilyl)-N-methyltrifluoroacetamide (TBDMS) and 50 µL of anhydrous acetonitrile at 70 °C for 30 min. The TBDMS-derivatives of amino acids were analyzed by gas chromatography coupled to a mass spectrometer (GC–MS). Acid hydrolysis led to conversion of glutamine and asparagine to glutamate and aspartate, respectively. Therefore, results given for aspartate and glutamate include asparagine and glutamine, respectively.

## Fatty acid preparation

For analysis of fatty acids, 2 mg of the bacterial sample (freeze-dried cell pellet) were suspended in 2 mL hexane using an ultrasonic bath for 5 min. The supernatant was transferred into another vial und dried under a stream of nitrogen. The residue was treated with a mixture of HCl (3 M) in methanol at 70 °C over night to form fatty acid methyl esters (FAMEs). The mixture was again dried under nitrogen, dissolved in hexane and analyzed by GC–MS.

## Gas chromatography-mass spectrometry (GC–MS) analysis

GC-MS analysis was performed with a QP2010 Plus gas chromatograph/ mass spectrometer (Shimadzu) equipped with a fused silica capillary column (Equity TM-5; 30 m, 0.25 mm, 0.25 µm film thickness; SUPELCO) and a quadrupole detector working with electron impact ionization at 70 eV. Aliquots (0.1 to 6 µL) of the derivatized samples were injected in 1:5 split mode at an interface temperature of 260 °C and a helium inlet pressure of 70 kPa. Selected ion monitoring (SIM) was used with a sampling rate of 0.5 s. LABSOLUTION software (Shimadzu) was used for data collection and analysis.

For the measurement of proteinogenic amino acids, the column was held at 150 °C for 3 min, followed by a temperature gradient of 10 °C min$^{-1}$ to a final temperature of 280 °C.

For the measurement of FAMEs, the column was kept at 100 °C for 2 min after sample injection. Following, the column was developed with a first gradient of 3 °C min$^{-1}$ until a final temperature of 234 °C. Subsequently, a second temperature gradient of 1 °C min$^{-1}$ until a final temperature of 237 °C, and a third temperature gradient of 3 °C min$^{-1}$ to a final temperature of 260 °C was performed.

The retention times (Rt) for the substances used are the following: Alanine (Rt = 6.0), Aspartate (Rt = 14.8), Glutamate (Rt = 16.3), C16:1 (Rt = 29.2), and C18:1 (Rt = 44.3).

## GC–MS- Data processing and isotopologue calculations

Three biological replicates were listed for each data series, each measured three times to create technical replicates. A data point is therefore the average of a total of nine measurements. In addition, the results of an "unlabeled sample" ($CO_2$ and naphthalene with only natural abundance of $^{13}C$, therefore containing approx. 98.9% $^{12}C$ and 1.1% $^{13}C$ for each carbon atom) were subtracted from each measurement, in order to eliminate the influence of the natural abundance of $^{13}C$. The results of our measurements therefore only show $^{13}C$ excess values.

The calculation was performed with an inhouse excel based macro using the solver function of excel. The calculation affords measurement of reference samples and measurement of the $^{13}C$-enriched samples.

Briefly, there are three main calculation steps using linear regression analysis[34,61,62]:

1. The contribution of the derivatization reagent is determined: An abundance matrix is constructed from theoretical values, for example for a molecule with three (for example alanine) C-atoms (each containing 1.11% $^{13}C$). As result vector measured values of the natural abundance standard are used. The calculated multiplication vector represents the contribution of the derivatization reagent.
2. These values are used for the construction of a second matrix. The result vector now contains measured values of the $^{13}C$ enriched sample. The multiplication vector now represents the relative abundance,

which after normalization corresponds to absolute enrichment (natural + artificial $^{13}$C abundance).

3. In a third step the relative abundance values form the result vector. The matrix is constructed as in step 1. The resulting multiplication vector, after normalization now represents $^{13}$C excess values.

The results from the excel based calculations have been validated by comparing it to the measurement of reference mixtures containing commercial $^{13}$C-isotopologues at known concentrations. The software package "ISOTOPO" is based on the same calculation principles[61] and can be downloaded free of charge from:

https://www.uni-wuerzburg.de/forschung/spp1316/bioinformatics/isotopo/

The freely available package is for any non-commercial use. The package contains three additional files: software executable setup (installer), one dataset file and one excel file (which can be used to convert data from excel to '.iso' format).

Isotopologue calculations of the amino acids were typically performed with molecular fragments at m/z of $[M - 57]^+$, where M is the molecular mass of the respective TBDMS-derivative.

In case of fatty acids McLafferty fragments were used with m/z = 74. An example of the formation of the different fragments is given in Fig. S9 and the re-arrangement of fatty acids to the typical McLafferty fragment used for fatty acid analysis is shown in Fig. S8.

These $^{13}$C excess values deriving from measurements are compared to the maximum theoretical value deriving from the $^{13}$C excess present in the culture medium.

The theoretical probability $p_k$ of obtaining a certain number of labels in a molecule can be described as a binomial distribution and is calculated using the following formula:

$$p_k = p^k \cdot (1-p)^{n-k} \cdot \binom{n}{k}$$

In our case, p describes the probability of incorporating a $^{13}$C atom, n is the number of carbon atoms in the molecule and k is the number of labeled carbon atoms. The term $p^k$ therefore describes the probability of the incorporation of labeled carbon atoms, $(1-p)^{n-k}$ the probability that no $^{13}$C atom will be incorporated in the other positions and $\binom{n}{k}$ expresses the various possible distributions of the label across the molecule.

The probability that exactly one of the atoms in an alanine molecule (M+1 species) in the experiment with $^{13}$C$_{10}$-naphthalene is labeled ($p = 0.039$; $n = 3$; $k = 1$) is therefore:

$$0.039^1 \cdot (1-0.039)^{3-1} \cdot \binom{3}{1}$$
$$= 0.039 \cdot (0.961)^2 \cdot 3$$
$$= 0.039 \cdot 0.924 \cdot 3$$
$$= 0.108 = 10.8\,\%$$

Accordingly, the probability that exactly one of the atoms in alanine (M+1) in the experiment with $^{13}$CO$_2$ is labeled ($p = 0.051$; $n = 3$; $k = 1$) is: 13.8%.

## Raman microscopy

For Raman microcopy, N47 was grown as described above in 50 mL culture volume in 100 mL serum bottles, some with natural abundant bicarbonate buffer and some with 50% $^{13}$C-labeled bicarbonate. After 6 weeks, the cultures were harvested and prepared for Raman microscopy.

To prepare cells for Raman analysis, the cell suspension was centrifuged for 10 min at $15,000 \times g$ in 50 mL centrifuge tubes (Sarstedt, Germany). The supernatant was discarded and the pellet was resuspended in

1 mL of phosphate-buffered saline (PBS, pH 7.2, 0.4 mM NaCl, 8 mM KH$_2$PO$_4$). The sample was then centrifuged again ($5675 \times g$, 10 min) and the pellet was collected and resuspended in 250 μL of paraformaldehyde (PFA, 4% (w/v)) and 750 μL of PBS. After 2 h of fixation the sample was washed twice with 500 μL of PBS. Finally, PBS was gradually replaced with 500 μL of ethanol (50, 80, 100%, v/v, respectively). The fixed cells were stored at −20 °C until analysis.

The Raman spectra of single cells were captured using an inVia™ confocal Raman microscope (Renishaw incorporation, New Mills, UK) equipped with a 532 nm excitation laser and coupled with a 100x N-PLAN, NA 0.85, WD 0.27 objective lens (Leica, Germany). The acquisition time was 10 s with 3 accumulations at a laser power of 5 mW. Mirrored stainless steel was used as substrate for mounting the samples, because the high reflectivity enhances the overall efficiency of the Raman signal collection. In addition, this support does not exhibit significant Raman-active vibrations in the spectral range.

The captured spectra were analyzed using the analysis software R[63] and imported using the hyperSpec package[64]. 20 spectra from $^{12}$C and $^{13}$C-labeled cells were captured from different locations of the sample to ensure reproducibility of the sample. K-mean cluster analysis was applied to separate the spectra of the cells labeled with $^{12}$C from $^{13}$C.

## Metagenome from N47 culture

We isolated again DNA according to the manufacturer's instructions (DNeasy PowerLyser Power Soil Kit, Qiagen GmbH, Germany) from a "fresh" N47 culture in March 2022 to reanalyze the complete metagenome with new tools and databases compared to Bergmann et al.[22]. Qubit fluorometric assay (Invitrogen, USA) was employed to determine DNA concentration, and 100 ng of the DNA sample was send to external company CeGaT GmbH (Tübingen, Germany) to be sequenced with the Illumina DNA Prep Library Preparation Kit, utilizing the NovaSeq 6000 Illumina platform with a sequencing length of 2 × 150 bp. Demultiplexing, adapter trimming, and quality trimming of the sequencing reads were conducted using Illumina bcl2fastq (v 2.20), Skewer (v 0.2.2), and BBMap (v 38.67), respectively. MEGAHIT (v 1.2.8)[65] was employed for the assembly of generated reads. Bowtie 2 (v 2.2.5)[66,67] and Samtool (v 1.9)[68] were used for indexing and sorting alignment. Binning was accomplished with MetaBAT 2 (v2.12.1)[69] using the parameters: minContig length of 1500 bp, minCV 1.0, minCVSum 1.0, maxP 95%, minS 60, and maxEdges 200. Bin quality was assessed using the lineage_wf command of checkM (v1.1.3)[70]. Protein-coding genes were annotated with the distilled and refined annotation of metabolism (DRAM) tool[71] without the installation of the UniRef90 database[72]. Metabolic pathways were predicted through KEGG[73] pathway profiling of DRAM annotations. The analysis resulted in three bins, whereof bin 1 could be identified as *Candidatus* Desulfobacterium strain N47 with 72.9% completeness and no contamination or strain heterogeneity, bin 3 was assigned to bacteria but calculated only to have 17.55% completeness. Bin 2 had 0% completeness and is most likely a binning artifact. Only the N47 bin was further analyzed.

The genome sequence of N47 was deposited in the NCBI GenBank, BioProject PRJNA1066605.

## KEGG database screening

To identify already known microorganisms with genomes encoding both the complete TCA cycle and WLP in their genomes, we made use of the Kyoto Encyclopedia of Genes and Genomes database (KEGG) (https://www.kegg.jp/)[74] by applying the "BRITE Hierarchy Files" on the "organisms in taxonomic ranks" (08611) database (last updated: July 16, 2023). Organisms in taxonomic ranks database was searched for organisms that encode the complete TCA pathway with module number M00009 and at the same time for the complete WLP with module number M00377. For evaluating the metabolisms completeness KEGG classification was considered for each microorganism. Search results from this study correspond to the database content from the 17.07.2023. Some other organisms were identified by literature research.

## Statistics and reproducibility

As the enrichment culture N47 is very slow growing, it is difficult to obtain enough cell material for extensive repetitions and replicates. Therefore, the cultures for the GC-MS measurements were grown in triplicates for each setting (with labeled bicarbonate / with labeled naphthalene controls with no labeled substrates), and each culture was analyzed in technical triplicates. For the measurements of fatty acids, the triplicates had to be pooled together to obtain enough cell mass for analysis, but samples were still measured in technical triplicates. The reproducibility of the GC-MS measurements was in the range of 0.0% and 2.5%, between the technical replicates.

Growth of the cultures was monitored by measurement of $CO_2$ evolution using the DeltaRay. Two samples were taken for each time point from each culture as technical replicates. The difference between the two replicates of each sample is in the range between 0.01% and 0.2%, with only one outlier with a difference of 1.9%. The deviation of the final results from this is between 0.02% and 0.44%, and 2.44% for the outlier. This has no significant effect on the interpretation of the results.

For the enzyme assays, large amounts of N47 (from two to three separate cultures) were harvested and a batch of cell-free extract was prepared, from which one assay with all controls could be performed. Each different assay condition was set up in triplicates, in three separate cuvettes but from the same cell-free extract. The assay volume was 1 mL. All samples were measured in parallel in the photometer. In Table 1, the mean value and the standard deviation for the three measurements is given. Calibrations were also prepared in three separate set-ups and measured in parallel.

## Reporting summary

Further information on research design is available in the Nature Portfolio Reporting Summary linked to this article.

## Data availability

All data are available in the main text or the supplementary materials. Source data for Tables 1 and 2 can be found in the supplementary data table. All other data are available upon reasonable request from the corresponding author. N47 genome was used from Bergmann et al.[22] and downloaded for reanalysis from GenBank (accession numbers FR695864-FR695880, Table S1) and reanalyzed via DRAM (Distilled and Refined Annotation of Metabolism) tool[71], without installation of UniRef90 database, metabolic pathways were predicted by KEGG[73] pathway profiling of DRAM annotations. All other data are available on the KEGG online database, last search date was the 17.07.23. A new metagenome from N47 culture from March 2022 was analyzed and uploaded to the GenBank. This Whole Genome Shotgun project has been deposited at GenBank under the accession JBAMMY000000000. The version described in this paper is version JBAMMY010000000.

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

## Acknowledgements
We thank Ivan Berg and Christopher Bräsen for helpful discussions and for critically reading the manuscript. We acknowledge funding from the Hans-Fischer-Gesellschaft 2023-CH to C.S. and C.H. as well as the German Research Foundation, DFG (grant EI 384/16-1) to C.S. and W.E.

## Author contributions
Conceptualization: R.M., W.E. and I.H. Methodology: R.M., I.H., W.E., C.H., C.S., L.V., I.E. and Y.K. Investigation: I.H., C.S., L.V., I.E., Y.K., F.G., and M.H. Funding acquisition: R.M., W.E., and C.H. Writing – original draft: I.H., C.S., and L.V. Writing – review & editing: I.H., C.S., L.V., C.H., R.M., and W.E.

## Funding

## Competing interests
The authors declare no competing interests.
