## [Transparent peer review file · Communications Biology]

Chemoorganoautotrophic lifestyle of the anaerobic enrichment culture N47 growing on naphthalene

Corresponding Author: Professor Rainer Meckenstock

This manuscript has been previously reviewed at another journal. This document only contains information relating to versions considered at Communications Biology.

Version 0:

Reviewer comments:

Reviewer #1

(Remarks to the Author)

The authors have done a good job improving the clarity of the presentation of these results.

I am very satisfied that the authors have addressed many of the questions I had previously in a former review.

The authors should be commended as this is a very difficult system to work with (slow growing organisms with an insoluble substrate).

I just have a few minor comments left:

Abstract

I don't think it is typical to have citations in an abstract?

Line 21-22 - "surprisingly" is used 2x. Delete

Main text: ANME not defined.

Lines 50-53 in (for reviews... etc) Why the parentheses? I would remove.

To my understanding the term chemoorganoautotroph has only previously been used for anaerobic methane oxidizers, correct? It is not a widely appreciated concept...

The use of e.g. or i.e. (lines 36, 38) instead of just saying "for example" is weak grammar and should be reworded if possible to improve readability.

Reviewer #2

(Remarks to the Author)

For this review, I have used the file 32362_0_related_ms_841759_sq4wk4.docx, unzipped from the Reviewer Zip Source File, which contains edits with track changes, and line numbering matching the rebuttal letter. I haven't checked the consistency of this file with the final manuscript, which I assume is alright.

In this revised version, the authors have largely addressed my concerns. The tenor has been largely toned down where results do not support definitive statements, which is fine. Also, the revised discussion on the impact of carboxylation and isotope exchange added to the much-needed critical evaluation of the results. As a side note to one of the author's replies, it was not the intention of this reviewer to suggest an overinterpretation of the data, but to stimulate a self-critical evaluation of the results – which I believe is justified when extraordinary claims are being made.

I do not have any further objections, save for a couple of scattered observations.

Lines 950-fwd: considering that the authors agree with my comment regarding naphthalene measurements in HMN, I wouldn't send out the message that it is impossible to measure naphthalene in HMN. This part could be rephrased to read '... measurements of naphthalene concentrations can only be achieved by withdrawing HMN aliquots <references>, which however will alter the mass balance and isotope ratios. We have, therefore, ...'. Or something to that effect. Figure 3: format ¹³C in figure panels in superscript. In the lower panels, replace 0% ¹³C with 1% ¹³C or natural abundance ¹³C. Panels C and D, explained in the legend, are not marked in the figure.

Reviewers' comments:

Reviewer #1 (Remarks to the Author):

The authors have provided detailed answers to all comments by the reviewers. I have no further questions.

Reviewer #2 (Remarks to the Author):

I still believe this work uncovered a very interesting labeling pattern, however the main interpretation and conclusion, and proposal of a new type of metabolism are in this reviewer's opinion not fully supported. The authors addressed some of the comments and added new data to the revised manuscript. The latter consisted mainly of Raman spectroscopy of the N47 culture, analysis which showed the presence of newly produced glycogen.

Raman spectroscopy is to my knowledge a largely qualitative method: it shows the presence of compounds, but it is difficult to evaluate their amount.

Reply: Raman spectroscopy is actually quantitative and can be used to assess concentrations if you have appropriate standards. We agree that this is quite difficult with some compounds in microbial cells.

Other missing details:

- the red shift should be shown (non-label versus label conditions). Figure 3 now shows only the spectra of the labeled culture. The non-labeled state (reference) is needed to have a clear impression of the shift.

Reply: As suggested by the reviewer, we now show the red-shift in the glycogen induced by ¹³C-carbon incorporation (new figure 3 B) and added an unlabeled spectrum (in Figure 3 A). As now explained in the text (lines 461 - 467), the label incorporation from labeled bicarbonate into amino acids is also shown by the red shift of the phenylalanine band.

- it is difficult to estimate if glycogen is produced in sufficient amounts to act as storage compound.

Reply: Yes, that is true. We can only qualitatively interpret the data here.

- a substantial amount of label is needed in the cells to make for clear Raman shifts, as a rule of thumb in the range of 10%. The new section mentions only the amount of label added to the media. How much label was assimilated by cells?

Reply: As pointed out above by the reviewer, it is difficult to do a quantification of the label incorporation into phenylalanine at this time as pointed out in the text (now lines 464 - 467).

While this is a nice addition, it doesn't resolve the hypothetical separation of catabolism and anabolism proposed as mechanism for the observed isotope effects.

Reply: The addition of the Raman spectra was not intended to show the separation of catabolism and anabolism but to answer the reviewer's question for storage compounds. This was solved since it clearly shows the presence of polyphosphate and glycogen as explained in the text in line 4453 - 461.

It remains unclear how the two processes could work independent of each other, as speculated, considering that there is a common acetate/acetyl-CoA pool in the cell. I think this remains one of the main issues with the interpretation and main conclusion of this work. There is no compartmentalization, excluding spatial separation. Evidence for storage compounds is not very convincing (see comments above). As for a temporal separation, this was not tested. Perhaps it could be tested in physiology assays: a temporal separation could manifest itself as a sort of stepwise mass balance mismatch, where naphthalene consumption is disconnected from sulfate reduction. It can be tested by following naphthalene consumption and sulfide production at close intervals (high temporal resolution). Consumption of naphthalene not matched out by stoichiometric sulfide

production, and accumulation of storage compounds could support this hypothesis.

Reply: I appreciate the ideas of the reviewer but I am afraid that this is not possible. The yield coefficient of such an anaerobic culture is in the range of less than 10 %, which is due to the extremely small energy gain which is now explained in line 575 - 585. Hence, differences in CO₂ production/sulphate reduction due to different fluxes into storage compounds would be very small and by far below the accuracy of any measurement. Furthermore, there will be no time steps of the metabolism measurable in a batch culture because cells are not synchronized and different cells perform different metabolic steps at the same time. Due to the extremely slow growth of these organisms an experimental synchronization seems impossible to me.

I also ask the reviewer not to overdo with the interpretations and the expectations for this manuscript. Here, we report on the exceptional finding that the whole biomass is built from CO₂ indicating separation of catabolism and anabolism. This is novel and complete enough to justify an excellent publication. The possible explanations were added to put our findings into the light of other known related phenomena. This doesn't mean that our organism is following such a strategy, but this will be subject to future work. I think this is clearly pointed out in the discussion.

The new genome information summarized at line 75-81 adds reasons to doubt the presence of a fully functional TCA, and as reviewer 1 observed, the low activity detected for 2-oxoglutarate OR could be contributed by the contaminant bacterium. The authors have not commented on this. A complete genome of N47 will remove doubts on the presence of the complete pathway.

Reply: Thank you for this valuable criticism! At the moment we do not have a complete (100 %) genome of the N47 strain available, but we show now that the absence of four subunit genes in the N47 metagenomic bin, despite the presence of other related subunits, is likely due to limitations in the bioinformatics pipeline rather than their complete absence in the organism's genome. A bin completeness of ~73 % suggests that some genes were missing during assembly or binning. Possible reasons for this could be assembly fragmentation where the genome may not have assembled into full contigs, causing some genes, including subunits, to remain unassembled or fragmented and thus excluded from the bin. Another reason could be low coverage, certain regions of the genome may have had low sequencing coverage, making it difficult for the pipeline to recover those genes during assembly or binning. We have reanalyzed the unbinned metagenome and could find all four genes encoding the missing subunits. These issues commonly occur in metagenomic analyses, especially with incomplete bins, and it is likely that the missing subunit genes were simply overlooked during the process. We have changed the text in the manuscript (lines 75f) and included the unbinned genes into Table S1 and added the contigs protein sequences to the supplementary materials.

As shown in table 1, the activity of 2-oxoglutarate OR is clearly in the same range of all other enzymes of the TCA which makes the argument of reviewer 1 obsolete. We do not want to make changes in the manuscript if this lowers the quality rather than increases. Since also the genomic information supports the presence of 2-oxoglutarate OR it does not really make sense to doubt that.

Lines 78f: "The aconitate hydratase, also known as aconitase, was identified only in the N47 metagenome but not in the proteome and some subunit genes (2-oxoglutarate:ferredoxin oxidoreductase subunit delta, succinate dehydrogenase flavoprotein subunit, succinate dehydrogenase iron-sulfur subunit, succinate dehydrogenase cytochrome b subunit) were not found in the metagenomic bin but in the unbinned metagenome and in the proteome. Additionally, all other subunits of these enzymes could be identified in the bin and their absence is likely due to limitations in the bioinformatics pipeline rather than their absence in the organism's genome. The bin's completeness of only 72.90 % and 0 % contamination, also suggests that genes were missed during assembly or binning (Table S1). Hence, both WLP and TCA are complete."

Other

L185 fwd. (according to word doc with track changes): I think the label dilution by the carboxylation reaction cannot be considered insignificant. After all, this is 0.9 out of 10%, or 9% of naphthalene-derived carbon.

Reply: Label incorporation by carboxylation might contribute to the overall labeling pattern, but it cannot be the sole source of the labeling pattern we observe here. This is now addressed in a new paragraph in lines 324 - 333: "Influence of naphthalene carboxylation: In a scenario where acetyl-CoA is made from naphthalene, one has to consider that the degradation pathway involves the carboxylation of naphthalene (C10 compound) to naphthoic acid (C11 compound), as the initial activation step. Therefore, the carboxyl group could be labeled to a maximum of 10 % in experiments with 10 % $^{13}\text{CO}_2$. Thus, only 10 mol% of one carbon atom in the resulting naphthoic acid could be labeled from $^{13}\text{CO}_2$ and the theoretical label incorporation into naphthalene-derived acetyl-CoA is only less than one percent ($\frac{1}{11} * 10\% = 0.9\%$), which is significantly smaller than the observed values for the acetyl-CoA precursor in fatty acids (i.e. 6 – 11 % M+1) (Table 2). Therefore, the ^{13}C -data observed in the experiment with $^{13}\text{CO}_2$ cannot be explained by the initial carboxylation reaction of naphthalene only."

The statement that it is impossible to measure naphthalene dissolved in HMN is incorrect: this was done before, by different methods: <https://www.ncbi.nlm.nih.gov/pubmed/11207761>, <https://www.ncbi.nlm.nih.gov/pubmed/18811643>

Reply: Thank you for pointing this out, we will consider it for future experiments! Unfortunately, the concentration cannot be measured anymore because there are no more samples from the cultures.

The manuscript has several very speculative statements like "(examples from discussion around L420): "the possible danger is ...", "they have probably not enough carbon source readily available ...", "have probably not enough carbon source readily available ..."

Reply: The sentences were rephrased to "A speculative scenario explaining the observations would be...", "In the extremely oligotrophic environments inhabited by these organisms, there might not be enough carbon source readily available...". The authors want to make clear that this is speculation based on the given results and cannot be proven at the moment, but still want to offer this possible explanation for the reader to consider. New lines: 559 fwd.

L245: "Hence, it can be excluded that naphthalene oxidation proceeds through the WLP ..." vs. L442, "... indicate the use of the WLP for oxidation of acetyl-CoA...".

Reply: There was a typo in former line 236 (now line 380) and the sentence was supposed to state "It can be concluded that...", then these two sentences both state the same thing. During our current revisions, this sentence was deleted.

Reviewer #3 (Remarks to the Author):

We appreciate the authors' work on this difficult study and acknowledge the potential significance of their findings. But I wonder if the presentation could be more balanced, given the challenges to interpret the data. It would not take away from the manuscript to show the data and just say that the isotope pattern is confusing, and points to the likelihood of "chemoorganoautotrophy". I think it would be more clear to define this term as autotrophic growth using an organic energy source.

Reply: Thanks for the suggestion: we reduced the stringency of the statement and changed sentences in the abstract and in the conclusion, for example:

lines 19 – 21: "Combining ^{13}C - labeled substrates and mass-specific GC-MS analysis of amino acids and fatty acids with enzyme activity assays suggests that N47..."

lines 599 - 601: "Therefore, the data point to the likelihood of N47 being the first example..."

To strengthen the claim of an unusual metabolism might actually require conducting the experiment again using 100% U-labelled naphthalene, although this is likely not possible. It would be crucial to demonstrate that there is no increase in the +2 species at the conclusion of the experiment. Currently, the 10% C13 naphthalene used is quite low, especially when spread over 60 days, and there is no characterization of the isotopologue profile of naphthalene before and after the experiment.

Reply: We do not see the necessity to use 100 % labeled naphthalene because the mass spectroscopy used is by far sensitive and accurate enough to quantify the incorporated carbon from naphthalene.

The stable isotope fractionation of naphthalene with culture N47 has been determined almost 20 years ago and is very small ¹. There is no chance that isotope fractionation would change the labeling patterns observed here because they are orders of magnitude higher than natural fractionation which can only be measured with specialized isotope ratio MS.

This reviewer is still unsure about the calculations of the labelled species profile. For instance, if each carbon atom has a 6.3% chance of being ¹³C, the +1 species for a 1-carbon compound would have a ~6% abundance. For a 2-carbon compound, it would be 12%, and for a 3-carbon compound like alanine, it would be 18%. Ensuring these calculations are accurate is essential for the credibility of the findings.

Reply: Yes, we agree and indeed one can see the systematic increase of the M+1 species correlating with the number of carbon atoms of the amino acid in table 2. We added calculations in the materials and methods section (line 1107f) and explain this in the main text in lines 261 – 268. The lower ¹³C excess in our measurements can be explained by residual unlabeled protein from the unlabeled precultures (in our setup we look at protein derived amino acids) and dilution due to the treatment of the medium with N₂/ unlabeled CO₂ prior to its use. New lines: 267 - 269.

Other specific comments are as follows

Line~ 45: “The only known exception to the rule that organisms feeding on organic compounds are chemoorganoheterotrophic are methylotrophic microorganisms that oxidize their C1-substrate (e.g. methane) to CO₂ and produce part of their biomass by CO₂ fixation³. Another part of the methane can be only partially oxidized to formaldehyde and also fueled into anabolism, for example via the serine pathway. This less energy efficient metabolism is needed because these organisms do not produce acetyl-CoA in the degradation pathway, which excludes using the energy-rich compound directly for anabolism.”

It would be helpful to clarify whether the authors are referring to ANME. If so, are they discussing the metabolism of archaea or the sulfate-reducing bacteria? Reference 3 suggests that sulfate-reducing bacteria are complete autotrophs, obtaining energy from sulfate reduction, electrons from archaea, and carbon from CO₂. It appears the authors may be referencing archaeal metabolism within ANME. This is a key point of the article on why organisms would want to use CO₂ over reduced carbon for biomass, and it is not clearly demonstrated.

Reply: Yes, this sentence refers to the ANME-1 archaea studied on reference 3, this was added to the sentence: “The only known exception to chemoorganoheterotrophic organisms feeding on organic compounds are methylotrophic microorganisms. Among these, ANME archaea oxidize their C1-substrate methane to CO₂ and produce their lipids by assimilation of inorganic carbon.” New lines: 42 - 46

Line ~90: “drive a reversed oxidative TCA cycle when high levels of CO₂ are present”

Earlier you use rTCA for reverse TCA cycle. Is this the same thing? Should you say drive a reversed TCA cycle....How can the reverse TCA cycle be oxidative ?

Reply: The reversed oxidative TCA cycle is not the same as the reductive TCA cycle. The roTCA uses the same enzymes as the oxidative TCA cycle, but this can be pushed to run in the reverse direction

under very special conditions. This is explained in the first sentence of the paragraph and important references are given, the name was now added there for clarity: "A few organisms can use this cycle also in the reductive direction under very special conditions, such as high CO₂ partial pressure and elevated levels of citrate synthase in the cell, though the enzyme activities in this direction are lower than in the oxidative direction. This pathway is referred to as the "reversed oxidative TCA cycle" (roTCA)." New lines: 93 - 96

We propose that N47 is using a reversed oxidative TCA cycle, this was corrected throughout the manuscript wherever it was not entirely clear.

Line ~97: "In this study, we conducted experiments to show if N47 employs a chemoorganoautotrophic metabolism degrading naphthalene to CO₂ for energy conservation and assimilating CO₂ for biomass formation."

This sounds like you wanted to show a result that you already had in mind, rather than doing an experiment to determine which way carbon flowed. This objective should be reworded to frame as a question rather than to prove a desired result.

Reply: The last paragraph of the introduction was rephrased to include the objective of the study and then follow up with the main conclusion: "In this study, we analyzed the carbon flow in culture N47 and determined which pathways are involved in its metabolism. We found indications for both an active TCA cycle as well as for a WLP. ¹³C-labeling experiments with N47 and enzyme activities in cell-free extracts of N47 suggested that N47 employs a chemoorganoautotrophic metabolism of (i) degrading naphthalene to CO₂ for energy conservation and (ii) assimilating CO₂ for biomass formation at the same time. ... ". New lines: 118 - 126

Line ~160: "All but two enzymes could also be measured in the reductive direction, including the key enzyme 2-oxoglutarate:ferredoxin oxidoreductase which, however, is generally assumed to be fully reversible 27-30."

If the reaction is assumed to be reversible, could you explain why it cannot be measured?

Reply: We could not find any assay in the literature that can measure this enzyme in the forward oxidative direction and could not develop an assay ourselves. However, this seems to be the case in all studies on this topic.

Line ~163: "Neither a gene for ATP-dependent citrate lyase (ACL) nor for citryl-CoA lyase (CCL) was identified in the genome."

In the context of the rTCA cycle, it would be valuable to mention that the synthesis of citrate is expected to proceed through ATP-citrate lyase, the reverse reaction of citrate synthase, or another pathway. Readers are not expected to have that knowledge.

Reply: An explanation was added at this point: "One of these enzymes would be a likely candidate for citrate synthesis in a putative rTCA, catalyzing the reverse of the citrate synthase reaction. Their absence does not necessarily exclude a functional rTCA, because there are other options possible, like the roTCA mentioned above. This cycle uses the citrate synthase in the reductive direction, but with much lower enzyme activities than in the oxidative direction. " New lines: 142 - 146

Line ~167: "Even though the enzyme activities appear very low, they are still comparable in range to other studied organisms, for example enzyme activities of 36 nmol min⁻¹ mg⁻¹ and 10 nmol min⁻¹ mg⁻¹ were reported for isocitrate dehydrogenase and aconitase, respectively, in cell extracts of *Aquifex pyrofilus* growing on H₂, CO₂, O₂ and thiosulfate 31."

The enzyme activities in the TCA cycle do not fall within the same range, and the connection between this variation and the conclusion of a closed TCA cycle could be clarified. The first three enzymes show activity in the 1~3 nmol/min/mg range, the fourth at 70 nmol/min/mg, and the last two at <1 nmol/min/mg.

Reply: This appears to be normal within other studies as well, with enzyme activities ranging over two or even three orders of magnitude. Furthermore, our data are very reasonable with respect to the extraordinary slow growth of the culture which already implies low specific activities.

This was added in the manuscript to explain the case and give references of studies with similar results: “The big range of activities among the enzymes of the TCA cycle seems striking, but this appears to be not uncommon, since enzyme activities of the TCA cycle can range over two orders of magnitude. This might be caused by assay conditions mimicking the natural conditions for some enzymes better than for others. Furthermore, the low specific activities measured for N47 reflect the extremely slow growth with doubling times of one to two weeks.” New lines: 155 - 160

Line ~180: Table vs. Figure S1

Figure S1 is a good addition and perhaps more effective than Table 1 for conveying this information. Perhaps you could show all the reagents though as some enzymes catalyze multiple steps and in some cases were assayed together.

Reply: Enzymes assayed together are marked with an asterisk in the figure. Showing all reagents would make the figure very confusing and overloaded and would, in our opinion, take away from the presentation of the actual results.

Line ~177: “An overview of the TCA and WLP with the measured enzyme activities can be found in Figure S1. The 10 to 50-fold higher enzyme activities of the WLP compared to TCA enzyme activities indicate that the WLP is used for catabolism and the TCA for anabolism.”

Could you clarify how the data is indicating that? Can it not mean that the organism just uses primarily the WLP for everything?

Reply: This is possible from just looking at the activity data. Hence, the sentence was corrected to not indicate anything that is not supported directly by this figure: “... The 10 to 50-fold higher enzyme activities of the WLP compared to TCA enzyme activities might indicate that the WLP is used for catabolism, i. e. acetyl-CoA oxidation.” (lines 166 – 168). Furthermore, we added a new sentence: “Nevertheless, it could theoretically be that strain N47 is using the WLP for both acetyl-CoA oxidation and CO₂ fixation. However, this would imply that the WLP is in absolute equilibrium which is unlikely due to the fact that the oxidation of acetyl-CoA is coupled to sulfate reduction.” (Now line 168 – 171)

Line ~265: Table 2, row 3-5, labelled naphthalene experiment

Given that most percentages are close to 100%, I assume these reflect the population of individual species. If so, alanine (3 carbons), aspartate (4 carbons), and glutamate (5 carbons) should have different natural isotopic profiles. The lack of ¹³C-enrichment suggests that the +1 signal comes from natural ¹³C abundance, so averaging these might not be appropriate.

Reply: We have deleted the misleading M+0 column. In our calculations, (as shown in materials and methods line 1068ff), natural abundance values are already subtracted. We always discuss ¹³C excess values; in Table 2, the labeled species of all three amino acids show low but significant ¹³C excess values.

Nevertheless, we agree that averaging these values might not be useful. We deleted these lines from Table 2.

However, this highlights the importance of the ¹³C-bicarbonate experiment, supporting the synthesis of amino acids through WLP.

Can you show the calculations for the predicted isotopologue distribution for the various scenarios?

Reply: Naphthalene: In lines 230 - 237 we already predicted formation of M+2 isotopologues from ¹³C labeled naphthalene; in lines 249 - 254 we added the calculation for a maximum ¹³C labeling for the ¹³C naphthalene experiment if a total degradation to CO₂ took place.

CO₂: In lines 261 - 268 we added the calculation for a maximum ¹³C labeling for the ¹³CO₂ experiment under the same assumptions.

Maximum values are probably not achieved due to the slow growth of the culture and therefore residual protein from the unlabeled precultures, due to the treatment of the culture medium with N₂/CO₂ prior to its use and through incomplete degradation of naphthalene.

Figure S2:

This is a useful figure, but it was not mentioned in the main text. Please refer to it.

Reply: Figure S2 is now Figure S1, and it is now mentioned in line 136 and also at the corresponding Materials and Methods section in line 1002.

Line ~265 (Fatty Acids): "For fatty acids, typical fragments with a mass of 74 (McLafferty fragments) were measured (see Figure S5)."

If the masses are identical, reporting the retention times of C16 and C18 fatty acids would be informative. Could you also explain the selection criteria for the C16 and C18 fatty acids? Why are these two?

Reply: The masses are not identical before fragmentation in the quadrupole MS, only the mass of the fragment used for calculations is the same. C16:1 and C18:1 were the most abundant fatty acids, which were seen in the GC-MS measurements. The retention times for fatty and amino acids used in the manuscript are now added in the material and method section in lines 1065 - 1066.

Line 275 (Glutamate Labeling): Figure 1: compound 7 & 8

Alphaketoglutarate/glutamate can also be labeled on the C-alpha side from naphthalene (7 & 8), meaning the carboxyl-carbon and the oxo-carbon should also be red-lined, which leads to the double red-line in succinate (9).

Clarifying this point would be helpful.

Reply: The double red line in succinate derives from the symmetry of the molecule. In a first TCA round glutamate/oxoglutarate are labeled only on C4 and C5. Only in a second run of the TCA, C1 (carboxyl group) and C2 (α -C-atom) would carry a ^{13}C label. This is not shown for better clarity of the figure.

Line 278: "Thick red lines indicate two adjacent ^{13}C -atoms (C2 body, two labeled carbon atoms in the same molecule). Blue dots indicate ^{13}C -atoms derived from $^{13}\text{CO}_2$ (C1 body, one labeled carbon atom)."

Could you clarify what is meant by "C1 body"?

Reply: To avoid confusion, the term "C1 body" was removed, it just indicated that this is a one-carbon-compound to be consistent with the "C2 body" in the line above. New line 346

Line ~305: "Hence, the formation of M+1 labeled acetyl-CoA and its downstream amino acid products provided evidence for acetyl-CoA formation via CO_2 -fixation and naphthalene oxidation primarily to CO_2 rather than use of naphthalene oxidation products for anabolism (see also Figure 1)."

Please explain more clearly how natural abundance was accounted for? It is so hard to follow.

Reply: The ^{13}C from natural abundance does not influence the conclusion of this paragraph. Natural ^{13}C abundance is subtracted throughout all results. This is explained in materials and methods (lines 1069 - 1074) but not at every paragraph because it makes the written text even harder to follow. Only in Fig. S3 absolute ^{13}C values in CO_2 are shown. For clarity, we added a sentence: Hence, the natural ^{13}C abundance is subtracted for all data." (Now lines 200 - 201)

Line ~313: "This indicates that most of the acetyl-CoA which is used for anabolism and synthesis of amino acids from pyruvate or intermediates of the TCA cycle comes from CO_2 fixation rather than from naphthalene degradation which agrees with the experiment with ^{13}C -labeled naphthalene above."

Could you clarify what is meant by "synthesis of amino acids from pyruvate or intermediates of the TCA cycle"?

Reply: This was just a typo, it should be "pyruvate or intermediates of the TCA"; it was corrected in the manuscript. New line: 273.

Line ~314: "This indicates that most of the acetyl-CoA which is used for anabolism and synthesis of amino acids from pyruvate or intermediates of the TCA cycle comes from CO₂ fixation rather than from naphthalene degradation which agrees with the experiment with ¹³C-labeled naphthalene above."

The authors are trying to say that in Table 2 row 6, column 7, the number is 9.1%, which is close to 6.3% of overall ¹³C abundance in the culture, that the labelling of carbon one by one rather than 2 at a time.

However, if the overall ¹³C abundance is 6.3%, alanine should have approximately 18% +1 species. As it contains 3 carbons, each has 6.3% chance of being labelled. And by that, each amino acid (C3-alanine, C4-aspartate and C5-glutamate) should each have a different isotopologue profile and cannot be averaged.

Reply: The question is similar to an earlier question and is answered in more detail there. Indeed, one can see this increase of the M+1 species correlating with the number of carbon atoms of the amino acids in table 2. We agree that averaging these values might not be useful and deleted these lines in table 2.

Line ~316: "Hence, it can be excluded that naphthalene oxidation proceeds through the WLP (higher enzyme activity according to Table 1), and the TCA is used to produce anabolism intermediates because the labeled amino acids should still show the label of two carbon atoms of the acetyl-CoA units produced from naphthalene degradation."

Could you further explain why the data suggests acetyl-CoA generated from naphthalene is not degraded through WLP? A figure might help clarify this concept. It's unclear what the authors propose happens to the acetyl-CoA generated from naphthalene.

Reply: This was actually a typo and should read: "...it can be concluded that naphthalene..." During our revisions, the sentence was removed anyway. Thanks for finding this typo.

Line ~345: "For example, mass fragments by loss of the carboxylic group (i.e. including C-1) can be observed for most amino acids (fragments M-85 and M-159). By comparison of the mass distributions in these fragments with the intact amino acid molecules, some information about the positional ¹³C-incorporation can be deduced."

It seems that the authors are comparing the ¹³C-incorporation rate of M-57 with M-85 and M-159. The raw data of these should be presented.

Reply: To simplify the manuscript, table 3 was deleted as the results are not necessary for the interpretation of the data. Figure S18 was deleted as it is not needed anymore.

Such that M-57 of alanine has 2.4% +1 (Table 2), and M-85 has <0.1% +1. Does that not mean all the ¹³C should be in C-alpha? How would you explain that?

Reply: Table 3 and all conclusions thereof were deleted, as positioning of the ¹³C label was unclear in some fragments.

Line ~350, Table 3:

The convention in Table 3 should be C-alpha or C-beta instead of C1/C2. From line 360, I understand that C1 refers to the carboxyl carbon, but what does C2 represent? C-alpha?

Reply: Table 3 was deleted.

Line ~359: "The experiment with ¹³C-labeled naphthalene showed a higher amount of label in C-2 (mean 1.4 % ±0.8) compared to C-1, even though the overall amount of ¹³C was lower. This ¹³C-distribution can be explained by the pathways of amino acid biosynthesis where C-1 is mainly derived from CO₂ whereas the remaining amino acid chain is built up from acetyl-CoA (see Figure 1)."

How can the atomic% be higher while the amount is lower? This could use some clarification with the raw data.

Reply: Table 3 and conclusions thereof were deleted.

Higher amount of label in C-alpha means carboxyl-C is derived from CO₂? And the authors are suggesting that naphthalene-derived acetyl-CoA enters anabolism over here, as C-alpha is labelled more?

Reply: Table 3 and conclusions thereof were deleted.

Line ~363: "These 13C-patterns are therefore another indication that CO₂ fixation occurs via the WLP route and/or the reverse TCA cycle in contrast to complete degradation of acetyl-CoA into CO₂ via an oxidative TCA cycle."

The above statements are not clear and thus was not able to understand this statement.

Reply: This whole paragraph was removed and rephrased elsewhere in the manuscript for clarity.

Line ~375: Table 2, row 7-9, column 4,5,7,8 "The 13C-composition was characterized by high M+1 fractions in both labeling experiments (Table 2) with 13CO₂. This finding provided further evidence that acetyl-CoA used for biosynthesis was mainly formed from CO₂. The detected minor M+2 fractions in fatty acids from the labeling experiment with 13C-naphthalene could reflect that, in fatty acid metabolism, the separation of anabolism (i.e. using acetyl-CoA from CO₂ fixation) and catabolism (i.e. using acetyl-CoA from 13C10-naphthalene) is not absolute."

The table is too far away from the discussion.

The authors are suggesting that there shouldn't be +2 species in fatty acids if acetyl-CoAs are synthesized from CO₂ fixation, but the reason is not explained.

Reply: An explanatory sentence was added: "If the acetyl-CoA is built from CO₂ with 10 atom percent ¹³C label, the probability of the M+2 fraction with two labeled carbon atoms is about 1 % through statistical coupling." Now lines 214 - 216

Line ~385: "Taken together, the labeling patterns in amino acids and fatty acids indicate that strain N47 mainly oxidizes naphthalene to CO₂ via acetyl-CoA and further oxidation via the WLP, and synthesizes acetyl-CoA for anabolism de novo from CO₂ via the rTCA. Hence, it displays a chemoorganotrophic metabolism."

I would recommend repeating the experiment with 100% U-labeled naphthalene. If no significant +2 species are detected, the results would be more convincing. The current 10% level is low, especially over a 60-day period.

Reply: For the sensitivity of the applied mass spectroscopy, it does not make a big difference if 100 % or 10 % labeled naphthalene was used. Nevertheless, we appreciate the comment and will consider applying 100 % label in controls of future experiments. In the current experiments, the 10 % label was necessary to measure the degradation of the naphthalene by CO₂ evolution with reverse stable isotope labeling.

Line ~393: "To investigate the presence of storage compounds, Raman spectroscopy was applied. There was no indication..."

It's unclear which sample was used for Raman analysis.

Reply: The sentence was changed to explain what samples were used: "To investigate the presence of storage compounds, Raman spectroscopy of single cells was applied to N47 cultures that were grown with bicarbonate buffer labeled with 50 atom% ¹³C or at natural ¹³C abundance." New lines: 449 - 451

Line ~403: Figure 3 "When N47 was grown with 13C-labeled bicarbonate (50 atom%, Figure 3) or with natural abundance of 13C (not shown), a red-shift was observed in the glycogen-associated Raman band from 854 cm⁻¹ to 849 cm⁻¹, which indicates that the storage compound glycogen is built from CO₂."

Was the peak split or completely shifted? If only 50% of the bicarbonate carbon is 13C, how can the peak shift completely? This point could be clearer.

The observed is consistent with the incorporation of heavier isotopes, such as ¹³C, into the glycogen structure. The 854 cm⁻¹ band is often associated with C-O-C glycosidic linkages. The shift from

854 cm^{-1} to 849 cm^{-1} suggests that ^{13}C is indeed affecting the vibrational energy of this bond. The 5 cm^{-1} decrease aligns with expectations for partial isotopic substitution, where the heavier isotope reduces the frequency of vibration. Because the shift is relatively small (5 cm^{-1}), it implies that the isotopic substitution is partial rather than complete. If all carbons were ^{13}C , a more significant shift could be expected, possibly around 20-30 cm^{-1} depending on the specific bond. In our case, since 50 % of the bicarbonate carbon is ^{13}C , only half of the glycogen's carbon atoms are likely ^{13}C , which results in this moderate red shift as a combination of ^{12}C and ^{13}C contributions. The absence of a clear band split but the presence of a single shifted band suggests that the ^{12}C and ^{13}C isotopes are well-distributed within the structure rather than being clustered. This may result in a more homogeneous distribution of mass within the glycogen, producing a singular band shift rather than two distinct peaks (which would be seen if there were segregated regions of ^{12}C and ^{13}C). The shift from 854 cm^{-1} to 849 cm^{-1} indeed indicates successful incorporation of ^{13}C into the glycogen structure, affecting its vibrational characteristics.

Line ~506: "Such a separation in time would also be possible in strain N47 but would require a thorough regulation, similar to yeast 57."

Could such regulatory genes be identified in N47? Considering yeast is a eukaryote, can this inference be made for N47?

Reply: No, we do not find such genes in the genome other than regular enzymes for cell division. However, we also do not imply that N47 does use one of the presented regulatory phenomena. We just present these to demonstrate that analogous phenomena do exist and that they make sense. Furthermore, we discuss possible ecological implications. We better explain this now in lines 554ff.

Line ~520: "The observation of glycogen as a storage compound in N47 that is created from CO_2 clearly supports this hypothesis. Although the cells would "waste" some of their energy for CO_2 fixation, the time shift of catabolism and anabolism might be a safer strategy in extremely oligotrophic environments."

Has this temporal separation been observed in culture maintenance? It seems the hypothesis is that N47 transforms naphthalene into CO_2 and uses the energy to fix CO_2 for biomass. This concept appears to be blurred between cell division and biomass synthesis.

Reply: It has not clearly been observed, but this is likely not possible. If there would be a temporal separation, it would probably not take place in every cell at the same time, so looking at a culture bottle with high number of cells would only show an "average" of all the processes taking place in the cells. Yeast is an exception here, because in chemostates (not in batch cultures) it brings all cells into the same cycle. For bacteria, we would need synchronized cells which, seems impossible to me with respect to the slow growth of the strain.

The suggestion is that N47 synthesizes glycogen using energy from naphthalene degradation, which is then used for new cell production. If so, there doesn't seem to be any advantage in breaking down acetyl-CoA and synthesizing acetyl-CoA simultaneously, which would be necessary for glycogen synthesis. A schematic figure illustrating the proposed glycogen synthesis pathway would be beneficial. If acetyl-CoA is involved, the reasoning needs further clarification.

Reply: Yes, that's indeed the major novelty of the paper which is also clearly stated. All other organisms would use the acetyl-CoA from naphthalene degradation to produce glycogen. Only these chemoorganoautotrophs synthesize the acetyl-CoA for anabolism (including glycogen) from CO_2 .

The synthesis of glycogen could be simply from acetyl-CoA in established pathways by carboxylation to pyruvate or phosphoenolpyruvate and following steps. Since we did not investigate the precise pathway, we are hesitating to be too specific to not present things that are not supported by our data. Nevertheless, we added a sentence to clarify: "Glycogen could be produced by known pathways from the acetyl-CoA produced in the roTCA.", now lines 565 - 566.

Line ~530: “As the growth yield in these organisms is very low, a lot of naphthalene has to be oxidized to conserve energy and accumulate storage compounds as either polyphosphate or glycogen. Only a minor portion of the CO₂ produced by naphthalene oxidation is needed for biomass production.”

Including the calculation for this would be helpful for readers.

Reply: To clarify we added a paragraph including the calculation in lines 575 – 585.

Reviewer #4 (Remarks to the Author):

Thank you for taking the time and participating in the review of our manuscript!

REVIEWERS' COMMENTS:

Reviewer #1 (Remarks to the Author):

I don't think it is typical to have citations in an abstract?

Line 21-22 - "surprisingly" is used 2x. Delete

Citations were deleted (now line 18).

ANME not defined.

ANME is now defined in line 44.

Lines 50-53 in (for reviews... etc) Why the parentheses? I would remove.

Parentheses were removed.

To my understanding the term chemoorganoautotroph has only previously been used for anaerobic methane oxidizers, correct? It is not a widely appreciated concept...

We think that this is the most fitting term to describe the metabolism of N47 and explain the concept thoroughly throughout the manuscript.

The use of e.g. or i.e. (lines 36, 38) instead of just saying "for example" is weak grammar and should be reworded if possible to improve readability.

Several occurrences of e.g. or i.e. were reworded, for example in line 36.

Reviewer #2 (Remarks to the Author):

Lines 950-fwd: considering that the authors agree with my comment regarding naphthalene measurements in HMN, I wouldn't send out the message that it is impossible to measure naphthalene in HMN. This part could be rephrased to read '... measurements of naphthalene concentrations can only be achieved by withdrawing HMN aliquots <references>, which however will alter the mass balance and isotope ratios. We have, therefore, ...'. Or something to that effect.

The passage about the naphthalene measurements was rephrased to: "Measuring naphthalene concentrations during the experiments is possible (ref), but requires withdrawal of further aliquots of the water and HMN phases for analysis, which was avoided here in order to not alter the mass balance in the culture bottles. Hence, we measured the CO₂-evolution which directly monitors the amount of naphthalene oxidized to CO₂.", it is now in lines 785ff.

Figure 3: format ¹³C in figure panels in superscript. In the lower panels, replace 0% ¹³C with 1% ¹³C or natural abundance ¹³C. Panels C and D, explained in the legend, are not marked in the figure.

Figure 3 was corrected as suggested by the reviewer.